# Understanding Compositionality in Data Embeddings

## Abstract

Embeddings are often difficult for humans to interpret, raising potential safety concerns. To address this, we analyze embeddings from different data structures such as words, sentences, and graphs, and interpret them in an understandable manner. This study investigates the algebraic relations, specifically additive, between pairs of vectors that represent entities known to be similar across a particular feature. To this end, we apply two methods: (1) Correlation-based Linearity Detection, which measures the correlation between known attributes of objects and their embeddings, and (2) Additive Compositionality Detection, which decomposes embeddings into an additive combination of vectors representing specific attributes. Embeddings are evaluated from various models, layers, and training stages to explore their capacity to encode compositional relationships. Sentence embeddings, for example, can be interpreted as the sum of underlying conceptual components. Similarly, word embeddings can be interpreted as capturing a combination of semantic and morphological information. Graph embeddings in recommender systems reflect the sum of a user's demographic attributes. In all three types of data, the relationships between structured entities are encoded as vector operations in embeddings, with a simple operation such as addition playing a central role in expressing compositionality. Code will be publicly available on GitHub upon acceptance.

## 1 Introduction

The representation of entities, concepts, and relations as vector embeddings is a foundational technique in machine learning (Mikolov et al., 2013a;b). Despite their broad success, embeddings often lack interpretability. One approach to improving interpretability involves examining the *compositionality* of embeddings: if embeddings can be understood as a function of known, interpretable representations, the information they encode becomes more accessible. Recent studies have explored compositionality in various contexts, including sentence embeddings (Hewitt & Manning, 2019), word embeddings (Mikolov et al., 2013a), and graph embeddings (Bose & Hamilton, 2019), as well as broader aspects of neural network structures related to interpretability (Lepori et al., 2023). In particular, sentence embeddings and graph embeddings have shown evidence of a particularly simple, *additive* form of compositionality (Xu et al., 2023; Guo et al., 2023), though this has been observed primarily in limited scenarios.

Word embeddings map words into continuous vector spaces based on their contextual relationships, capturing semantic analogies like *king - man + woman ≈ queen* (Mikolov et al., 2013b). Sentence embedding, performed using neural networks like BERT (Devlin et al., 2018), GPT (Brown, 2020) or Llama (Touvron et al., 2023), extends this idea by creating embeddings for entire sentences rather than individual words. It is again reasonable to ask if they display the compositional property. In this case the analogue of *king - man + woman ≈ queen* would be *Cat ate mouse - mouse + bird ≈ cat ate bird*. Similarly, an interesting question is how graph embeddings capture relationships, like user preferences in bipartite graphs, and encode demographic features such as age or gender. More generally, embeddings represent objects through a relationship function or measurable features, defining a vector space that encodes relational or attribute-based information.

**Motivation** Understanding whether embedding vectors can be decomposed into distinct semantic components is a fundamental challenge for interpretability. If these embeddings can be broken down into parts

corresponding to linguistic inflection, word composition, or collections of attributes, then simple operations such as vector addition may reliably reveal underlying semantic or structural changes. While extensive research has documented additive properties in word embeddings, less is known about whether sentence and graph embeddings exhibit similar behavior. Our work addresses this gap by investigating the extent to which vector addition reflects genuine semantic composition, both in final embeddings and across the internal structure of Transformer models.

**Our Contributions**  Although the decomposition methods we use were introduced in previous work (Xu et al., 2023; Guo et al., 2023), our current study makes several novel contributions:

- Novel Analysis of Linearity: We rigorously assess how much of the semantic and structural information in embeddings can be explained by additive, linear components. Our analysis quantifies the degree of linearity present in embedding spaces, providing insight into how interpretable features emerge from models that are otherwise highly non-linear.

- Extensive Cross-Domain Experiments: We evaluate a diverse array of embeddings—including static word embeddings (e.g., Word2Vec), sentence embeddings from Transformer-based models (e.g., BERT, GPT, Llama), and knowledge graph embeddings used in recommender systems. Our experiments cover multiple model layers and training stages, revealing how the strength of additive signals varies with model complexity and over time.

- Quantitative Findings: Our results demonstrate, for example, that the strength of additive signals in SBERT embeddings increases by up to 15% in mid-level layers before declining in upper layers where task-specific representations dominate. For graph embeddings, the correlation between user behavior and demographic information at later training stages (300 epochs) is 1.5 times higher than at earlier stages (5 epochs), with both stages significantly outperforming the random baseline. Additionally, our analysis shows that word embeddings can be decomposed into distinct morphological and semantic subcomponents, and that graph embeddings encode user attributes via additive operations.

Building on methods from (Xu et al., 2023) and (Guo et al., 2023), we examine the extent of additive compositionality across a range of embedding types, including sentence, word, and graph embeddings, as well as those derived from large language models. For sentences, we focus on pretrained embeddings from Transformer-based architectures, namely GPT (Brown, 2020), Llama 2 (Touvron et al., 2023), and various BERT models (Reimers & Gurevych, 2019; Sellam et al., 2022). To investigate additive decomposability, we introduce a task in which a sentence, for example, *"Can you find me an Adventure movie playing at AMC NewPark in Newark?"*, is split into its constituent concepts (*location*, *theater_name*, and *genre*). We then assess whether these concepts can be additively composed within the embedding space, evaluating performance across different models, various layers within SBERT, and multiple training stages of BERT.

For words, we analyze pretrained word2vec embeddings (Mikolov et al., 2013b), a static model that remains widely used and offers a contrast to the Transformer-based approach. Word2vec embeddings are well known to decompose morphologically. In this paper, the extent to which this decomposition holds across multiple suffixes is investigated, as shown in the example: *weightlessness - less - ness + y + ly = weightily*. Guo et al. (2023) showed that knowledge graph embeddings trained on the MovieLens dataset (Harper & Konstan, 2015) could be decomposed into demographic attributes, even without this information being used in training. In this paper, we assess the robustness of this finding across an alternative embedding method, and examine the compositionality of embeddings across training stages.

**Findings**  Our experiments show that simple vector operations capture meaningful relationships across various embedding types. For example, we find that sentence embeddings from SBERT, GPT, and Llama-2-7B can be decomposed into additive components that correspond to core conceptual elements, with additive signals strengthening by up to 15% in mid-level layers before diminishing in higher layers. In the case of word embeddings, models such as Word2Vec can be decomposed into coherent morphological and semantic units, aligning well with external benchmarks like WordNet. Similarly, knowledge graph embeddings trained by

different scoring functions on the MovieLens dataset show that demographic attributes such as age and gender are encoded through additive operations, even without explicit attribute labels. These results demonstrate that, despite their non-linear origins, large-scale models retain a quantifiable degree of additive compositional structure, providing promising avenues for enhanced interpretability in representation learning.

## 1.1 Background and Related Work

The principle of compositionality—that the meaning of a whole can be derived from its parts—has long been central to linguistic theory and distributional semantics. The extent to which embeddings are compositional has been an area of research across multiple domains. Identifying the components of an embedding provides insight into the factors affecting its representation, as does understanding the rules by which those parts are combined.

Unlike earlier studies that focus on a single domain or merely report statistical significance, our work provides a unified, quantitative framework for analyzing additive compositionality across words, sentences, and graphs. By clearly separating our empirical findings from established methods, we offer both a rigorous methodological foundation and new insights into the linear signals within modern embedding spaces.

**Compositionality in Sentence Embeddings**  Research on compositionality in sentence embeddings has examined how models represent lexical and syntactic structures. BERT (Devlin et al., 2018) is not given explicit syntactic trees during training, however its representations capture significant syntactic information (Hewitt & Manning, 2019). Ettinger et al. (2016) and Dasgupta et al. (2018) explore semantic roles, scope, and natural language inference by modifying sentence structures, while Adi et al. (2016) evaluate embeddings using tasks like sentence length and word order. Probing tasks and representational similarity analysis (RSA) have further analyzed how models encode linguistic features, with studies like Lepori & McCoy (2020), Chrupała & Alishahi (2019), and Tenney et al. (2019b) investigating syntax, semantics, and sentence structure in models such as BERT and ELMo. RNNs and Transformers have also been shown to encode symbolic structures effectively (Soulos et al., 2020; Yu & Ettinger, 2020). Xu et al. (2023) show that sentence representations from SBERT, using the [CLS] token, exhibit additive compositionality. In this paper, we extend their findings to more recent language models.

**Compositionality in Word Embeddings**  Numerous approaches have been proposed to understand compositionality in word embeddings. Mikolov et al. (2013a) demonstrated that words can be decomposed semantically and morphologically, for example, *quickly - quick + slow = slowly*. Disentangled Representation Learning (Bengio et al., 2013) separates attributes in embeddings, enhancing interpretability by associating latent dimensions with discrete features. Additive compositionality in skip-gram word vectors has been theoretically justified (Gittens et al., 2017), with subsequent work relaxing these assumptions and proposing models that satisfy more realistic conditions (Seonwoo et al., 2019). Shwartz & Dagan (2019) examined compositionality via six tasks, exploring semantic drift and implicit meaning, while Andreas (2019) proposed a metric based on fidelity in reconstructing representations from primitives. In this paper, we use techniques from Xu et al. (2023) originally designed for the analysis of sentence embeddings, and show that the same techniques can be applied to word embeddings. We show that word embeddings can be decomposed into multiple prefixes and suffixes, extending the analysis from Mikolov et al. (2013b). We further compare word2vec embeddings with WordNet embeddings (Miller, 1995) to evaluate their semantic compositionality.

**Compositionality in Graph Embeddings**  While compositionality has been extensively studied in linguistics, less attention has been given to graph embeddings. Graph embeddings, such as those in knowledge graphs and recommender systems, encode relationships and attributes as constituent components within their vector representations. For instance, embeddings trained from movie ratings capture compositional information, including user preferences and movie characteristics. Recent work has explored how graph embeddings encode such components during training, with methods like adversarial loss used to isolate specific attributes (Bose & Hamilton, 2019; Fisher et al., 2020). In this work, we evaluate compositionality in graph embeddings across different scoring functions and training stages, focusing on how embeddings encode and combine diverse attributes and relationships.

**Compositionality in Deep Learning More Generally**  The ability of neural networks to reason compositionally has been studied across various architectures. Kim & Linzen (2019) compared Gated Recurrent Units (GRUs) and Simple Recurrent Networks (SRNs) for compositional generalization, while Wu et al. (2020) analyzed contextual word representation models across architectures. Research increasingly focuses on improving generalization to unseen compositions, with Lake & Baroni (2018) and Kim & Linzen (2020) highlighting limitations of RNNs, LSTMs, and Transformers in systematic tasks. To address this, Lake & Baroni (2023) proposed a meta-learning approach for systematic generalization. Hupkes et al. (2020) introduced task-independent tests to evaluate compositional generalization, showing differences across recurrent, convolutional, and transformer models. Additionally, structural and functional studies (Mu & Andreas, 2020; Lepori et al., 2023) explore how architectures contribute to or hinder compositional understanding.

## 2 Data Embedding and Compositionality Detection Methods

### 2.1 Embedding Techniques

**Embedding Words**  Word embeddings involve building vectors for words based on the distributional hypothesis: words that occur in similar contexts have similar meanings. Word embeddings can be divided into contextual (Peters et al., 2018; Devlin et al., 2018) and static (Mikolov et al., 2013b; Pennington et al., 2014) embeddings. Static embeddings model word meanings as one static vector, whereas the embeddings produced by contextual embedding methods differ based on the context in which the word occurs. In this paper we focus on the semantics and morphology of embeddings. Since contextual embeddings naturally additively encode morphology through character level or subword embeddings, we focus instead on static embeddings, specifically pre-trained skip-gram word2vec (Mikolov et al., 2013a).

**Embedding Sentences**  We focus on Transformer-based language models and examine the additive compositionality present in pretrained embeddings. Xu et al. (2023) show that sentence embeddings modelled by the [CLS] token of SBERT can be additively decomposed into the words in the sentence. We extend their results to examine compositionality through different layers of SBERT and through training stages of the MultiBERTs (Sellam et al., 2022). We further examine the additive compositionality present in sentence embeddings from GPT Embeddings and Llama 2. Since these are produced by averaging the word embeddings, we alter the task to instead show that sentence embeddings can be decomposed into a sum of the concepts contained within them. Details are given in Section 3.

**Embedding Knowledge Graphs**  Knowledge Graph Embeddings (KGEs) represent entities and relationships as vectors. Scoring functions over these vectors encode the graph's topology (Nickel et al., 2011). Two main scoring functions are widely used: multiplicative scoring (Yang et al., 2014), which models interactions through the product of entity and relation embeddings, and additive scoring (Bordes et al., 2013), where relationships are modelled as translations in the embedding space. These embeddings are evaluated on link prediction tasks. Guo et al. (2023) previously showed that KGEs trained on the MovieLens dataset using the multiplicative scoring function could be additively decomposed into attribute embeddings encoding information such as age or gender. In this paper, we examine whether KGEs trained using the additive scoring function exhibit similar compositionality. We further examine how this compositionality changes over training stages.

### 2.2 Compositionality Detection Methods

Previous work has explored methods to detect compositionality in embeddings by examining the relationships between entity attributes and their representations. Two approaches are *Correlation-Based Compositionality Detection* and *Additive Compositionality Detection* (Xu et al., 2023; Guo et al., 2023), which we describe briefly here. For each entity—whether user, word, or sentence—we consider two representations:

- A *binary attribute matrix* **A** (e.g., demographics or syntactic properties), where each row corresponds to an entity and each column represents an attribute. Entries $a_{ij} \in \{0, 1\}$ indicate the absence or presence of attribute $j$ in entity $i$.

- A *continuous embedding matrix* $\mathbf{U}$ (e.g., user behaviour embeddings or word embeddings), where each row corresponds to an entity's embedding and each column represents a dimension of the embedding space.

**Correlation-Based Linearity Detection**  This method uses *Canonical Correlation Analysis (CCA)* to uncover latent correlations between the attributes and embedding dimensions (Shawe-Taylor et al., 2004). Given the binary attribute matrix $\mathbf{A} \in \{0,1\}^{q \times n}$ and the embedding matrix $\mathbf{U} \in \mathbb{R}^{q \times m}$, where $q$ is the number of entities, $n$ is the number of attributes, and $m$ is the dimensionality of the embeddings, the goal is to find transformation matrices $\mathbf{W}_A \in \mathbb{R}^{n \times k}$ and $\mathbf{W}_U \in \mathbb{R}^{m \times k}$ that maximize the correlation between the projected data:

$$\rho = \max_{\mathbf{W}_A, \mathbf{W}_U} \mathrm{corr}\left(\mathbf{A}\mathbf{W}_A, \mathbf{U}\mathbf{W}_U\right)$$

Here, corr denotes the Pearson Correlation Coefficient (PCC) between the projected representations, and $k$ is the number of canonical components. The transformations $\mathbf{A}\mathbf{W}_A$ and $\mathbf{U}\mathbf{W}_U$ capture the most correlated aspects of the attributes and embeddings, respectively. We provide a detailed formal exposition of all settings necessary to define a CCA component, including the precise definition, the underlying vector space and mapping of semantic attributes, as well as illustrative examples. A complete derivation and examples are included in the Appendix A.

**Additive Compositionality Detection**  The additive compositionality detection method quantifies how much each attribute influences the embeddings (Xu et al., 2023). It assumes that an entity's embedding can be approximated as a linear combination of attribute embeddings. Given $\mathbf{A}$ and $\mathbf{U}$, the goal is to solve the linear system $\mathbf{A}\mathbf{X} = \mathbf{U}$ where $\mathbf{X} \in \mathbb{R}^{n \times m}$ is the matrix of attribute embeddings to be learned.

A *leave-one-out (LOO) experiment* is performed for each entity $i$. We firstly exclude the $i$-th row from $\mathbf{A}$ and $\mathbf{U}$ to obtain $\mathbf{A}_{-i}$ and $\mathbf{U}_{-i}$, then solve $\mathbf{A}_{-i}\mathbf{X} = \mathbf{U}_{-i}$ using the pseudo-inverse to obtain $\mathbf{X}$. We then estimate the left-out embedding using $\hat{\mathbf{u}}_i = \mathbf{a}_i\mathbf{X}$, where $\mathbf{a}_i$ is the $i$-th row of $\mathbf{A}$, and finally compare $\hat{\mathbf{u}}_i$ with the actual embedding $\mathbf{u}_i$ using (1) L2 loss, (2) cosine similarity, and (3) retrieval accuracy. More details are given in Appendix B.

**Hypothesis Testing**  To determine statistical significance, a non-parametric hypothesis test is performed by directly estimating the $p$-value through Monte Carlo sampling. First, the test statistic $T_{\mathrm{real}}$ is computed for the real pairing: for CCA, this is the canonical correlation $\rho_{\mathrm{real}}$; for additive compositionality, they are L2 loss, cosine similarity and retrieval accuracy based on the real pairing. Next, permuted pairings are generated by randomly shuffling the rows of $\mathbf{A}$ to disrupt the alignment, resulting in permuted datasets $\{\mathbf{A}^{(1)}, \mathbf{A}^{(2)}, \ldots, \mathbf{A}^{(N)}\}$.

For each permuted pairing $j$, the test statistic $T_{\mathrm{perm}}^{(j)}$ is calculated, and the $p$-value is estimated as $p = \frac{1}{N}\sum_{j=1}^{N}\mathbb{I}(T_{\mathrm{perm}}^{(j)} \geq T_{\mathrm{real}})$, where $\mathbb{I}(\cdot)$ is the indicator function. A low $p$-value indicates that the observed statistic is unlikely under random pairings, supporting the statistical significance of the real pairing.

## 3  Evaluating Compositionality of Data Embeddings

### 3.1  Sentence Embeddings

We evaluate the extent to which sentence embeddings from SBERT, GPT, and Llama can be additively decomposed into the concepts expressed in the sentence, using a dataset derived from the Schema-Guided Dialogue (SGD) dataset (Rastogi et al., 2020)[1]. We firstly look at the compositionality of sentence embeddings from the final layer of each model. We go on to examine the compositionality of sentence embeddings through different layers of SBERT, and finally look at how compositionality develops during the training stages of the MultiBERTs.

---

[1]https://github.com/google-research-datasets/dstc8-schema-guided-dialogue

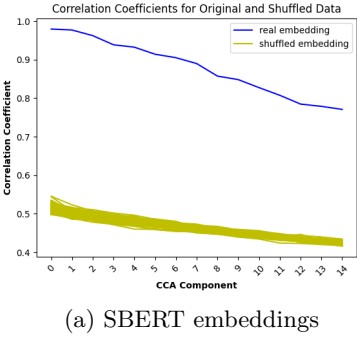
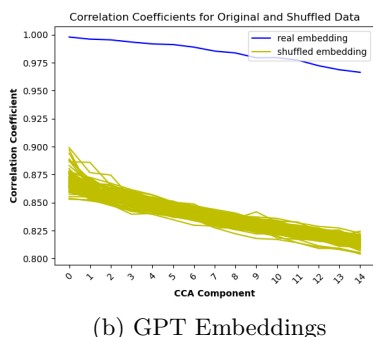

|                      |                      |
|:--------------------:|:--------------------:|
| (a) SBERT embeddings | (b) GPT Embeddings   |

Figure 1: Pearson's correlation coefficient (PCC) for true sentence-concept pairings and 100 permuted pairings. For both SBERT (Figure 1a) and GPT (Figure 1b) embeddings, we see that the correlation of true pairs is significantly higher than for random pairings.

### 3.1.1  Dataset

We build a dataset consisting of sentences annotated with concepts, taken from the Schema-Guided Dialogue (SGD) dataset. An example pair of sentence and concept labels is:

**Sentence:** Can you find me an Adventure movie playing at AMC NewPark in Newark?

**Concepts:** [*location*, *theater name*, *genre*].

We select 2,458 sentences, each annotated with minimum 3 and maximum 4 concepts from a total set of 47 concepts. The mean number of concepts per sentence is 3.16. There are 90 unique combinations of concepts used. The sentences used comprise the test set of the Schema-Guided Dialogue (SGD) dataset. This split was chosen because it is annotated with denser labels, providing more comprehensive concept coverage.

### 3.1.2  Experiments and Results

**Final Layer Embeddings**  We generate embeddings using pretrained sentence models, producing matrices $\mathbf{U} \in \mathbb{R}^{2,458 \times d}$, where $d$ is the dimensionality of the model. For SBERT[2] and Llama[3], embeddings are generated by mean pooling over the token embeddings, excluding padding tokens. For GPT[4], sentence embeddings are obtained from the OpenAI API, which provides precomputed embeddings representing the entire sentence. Each row of the matrix $\mathbf{U}$ consists of a sentence embedding for the corresponding sentence. We construct attribute matrices $\mathbf{A} \in \{0,1\}^{2,458 \times 47}$ with each row being a binary vector indexing the relevant concepts for the corresponding sentence.

We assess the compositionality of the sentence embeddings using the methods described in Section 2.2, and the $\mathbf{U}$ and $\mathbf{A}$ matrices described above. For the correlation-based linearity detection experiment, all sentences are used to compute correlations. For the additive compositionality experiments, sentences are grouped by their concept combinations, and mean embeddings are computed for each group.

For SBERT and GPT embeddings[5], we apply correlation-based compositionality detection to assess whether sentence embeddings correlate with their binary concept vectors. Figure 1 shows significant difference in correlation scores between real and permuted pairings (p-value < 0.01) for both GPT and SBERT embeddings, indicating that sentence embeddings are correlated with their binary concept vectors.

Results from the additive compositionality detection experiment are reported in Table 1. Cosine similarity and retrieval accuracy for the true pairings are all significantly and substantially higher than for the permuted pairings. The behaviour of SBERT and GPT embeddings are very similar, whereas Llama embeddings

---

[2]sentence-transformers/all-MiniLM-L6-v2
[3]meta-llama/Llama-2-7b-hf
[4]text-embedding-3-small
[5]We do not assess correlation for Llama embeddings as the dimensionality of these embeddings is too high

have a less substantial increase in compositionality over permuted pairings, as well as a lower retrieval accuracy (Hits@5). Plots of the distribution of the metric values for permuted pairs vs. real pairs for SBERT embeddings are given in Figure 2; the corresponding error bar plots, which summarize the distribution with the mean ± standard error, are shown in Appendix F. Results for GPT and Llama embedding are provided in Appendix D, Figures 9 and 10.

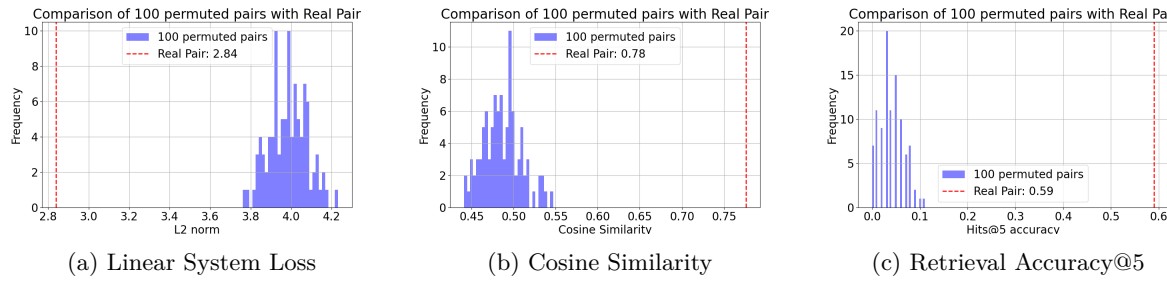

(a) Linear System Loss        (b) Cosine Similarity        (c) Retrieval Accuracy@5

Figure 2: Test statistics for SBERT's embedding decomposition. Dashed line is the average performance of $\hat{\mathbf{u}}$ learned from the sentence embedding. Bars are the distribution of the results from 100 random permutations. (a) L2 loss, (b) Cosine Similarity, and (c) Retrieval Accuracy@5 compare real SBERT embedding pairs to permuted pairs.

Table 1: Additive compositionality metrics for SBERT, GPT, and Llama embeddings. Figures in **Real** columns are the mean across all leave-one out experiments. Figures in **Permuted** columns are the mean across 100 permutations of the sentence-concept pairs. All increases of Real over Permuted values are significant with $p<0.01$.

| Embedding | Cosine Real | Cosine Permuted | Hits@5 Real | Hits@5 Permuted |
|---|---|---|---|---|
| **SBERT** | 0.7761 | 0.4865 | 0.59 | 0.0405 |
| **GPT** | 0.7753 | 0.4941 | 0.57 | 0.0399 |
| **Llama** | 0.9355 | 0.8153 | 0.52 | 0.0468 |

**Comparison between Different Layers**    We assess the additive compositionality of sentence embeddings derived from each layer of SBERT. At each layer of SBERT, the binary attribute matrix $\mathbf{A}$ remains the same as in the final layer experiment. Separate continuous embedding matrices $\mathbf{U}_i$ are built for each layer $i$. To generate sentence embeddings at each layer, we extract hidden states for all layers from the SBERT model and apply the same pooling method as used for the last layer (mean pooling over token embeddings). Each embedding is normalized to length 1.

The results, reported in Table 2, show that compositionality increases through the layers, peaking at layer 4 or 5. However, an abrupt drop in compositionality, as measured by cosine similarity, is observed at the last layer. This is in line with the phenomenon that semantic information is better encoded at earlier layers in the model, see for example experiments in the original BERT paper (Devlin et al., 2018). The fact that the cosine metric is fairly high for the random baseline. However, the discrimination between sentence meanings is still high as can be seen by fact that the Hits@5 for real pairings is consistently substantially higher than that for for permuted pairings.

**Comparison between Different Training Stages**    We further compare additive compositionality across different training stages of the MultiBERTs (Sellam et al., 2022). We again use the same binary attribute matrix $\mathbf{A}$ at each training stage. We build continuous embedding matrices $\mathbf{U}_{steps}$ for 0, 20k, 40k, 100k, 1000k, and 2000k training steps. We use the [CLS] token to represent sentences. At 0k steps, there is no significant differences between the values of the cosine similarity metrics for real and permuted despite high overall values (0.9884). This indicates that the embeddings lack differentiation and do not capture conceptual relationships. After 20k training steps, compositionality metrics significantly improve (Cosine Rel. Diff in

Table 2: Additive compositionality metrics across SBERT layers. Figures in **Real** columns are the mean across all leave-one out experiments. Figures in **Permuted** columns are the mean across 100 permutations of the sentence-concept pairs. All increases of Real over Permuted values are significant with $p<0.01$.

| Layer | Cosine Real | Cosine Permuted | Hits@5 Real | Hits@5 Permuted |
|---|---|---|---|---|
| 0 | 0.8889 | 0.7808 | 0.59 | 0.0397 |
| 1 | 0.9366 | 0.8671 | 0.57 | 0.0406 |
| 2 | 0.9397 | 0.8576 | 0.61 | 0.0390 |
| 3 | 0.9403 | 0.8330 | 0.64 | 0.0412 |
| 4 | 0.9408 | 0.8298 | 0.66 | 0.0400 |
| 5 | 0.9409 | 0.8273 | 0.62 | 0.0417 |
| 6 | 0.7761 | 0.4865 | 0.59 | 0.0405 |

Table 3), implying that additive compositionality emerges from training. Early stages (0k to 20k steps) show gains in cosine similarity and retrieval accuracy, demonstrating the model's ability to represent sentences as combinations of concepts. However, further training produces diminishing returns, as the model may shift focus to other linguistic features or risk overfitting. See Table 3 for details.

Table 3: Additive Compositionality Metrics at Different Training Steps of BERT. Number of training steps is given by the suffix to the model, e.g. cls_20k indicates the performance of the model after 20k training steps. Rel. Diff expresses this difference as a percentage of the permuted similarity, showing proportional improvement. Figures in Real columns are the mean across all leave-one out experiments. Figures in Permuted columns are the mean across 100 permutations of the sentence-concept pairs.

| Model | Cosine Real | Cosine Permuted | Cosine Rel. Diff | Hits@5 Real | Hits@5 Permuted |
|---|---|---|---|---|---|
| **cls_0k** | 0.9884 | 0.9882 | 0.02% | 0.44 | 0.0418 |
| **cls_20k** | 0.8787 | 0.7722 | 13.79% | 0.55 | 0.0407 |
| **cls_40k** | 0.8773 | 0.7724 | 13.59% | 0.48 | 0.0405 |
| **cls_100k** | 0.9201 | 0.8323 | 10.54% | 0.55 | 0.0417 |
| **cls_1000k** | 0.9545 | 0.9149 | 4.33% | 0.44 | 0.0408 |
| **cls_2000k** | 0.9538 | 0.9094 | 4.89% | 0.48 | 0.0415 |

## 3.2 Knowledge Graph Embedding

In Guo et al. (2023), it was shown that knowledge graph embeddings trained using the multiplicative scoring function (Yang et al., 2014) exhibit correlation-based linearity and additive compositionality. In this section, we firstly show that these results also hold for embeddings trained using the additive scoring function (equation 4, Bordes et al. (2013)). We go on to examine the evolution of correlation-based linearity over training stages, and finally examine the association between canonical variables and user attributes.

### 3.2.1 Compositionality across Diverse KGEs

We use the MovieLens 1M dataset (Harper & Konstan, 2015) with 6040 users, 3900 movies, and 1 million ratings (1–5). As in the sentence embedding experiment, we pair each entity, i.e. user, with two descriptions: a binary vector representing demographic attributes (gender, age and occupation) and a user embedding learned from movie preferences.

Continuous embeddings are trained as follows. We encode $(user, rating, movie)$ as relational triples $(h, r, t)$ and train 50-dimensional embeddings with OpenKE (Han et al., 2018), using the additive scoring function (equation 4), for 300 training steps. We assess the quality of the embeddings on a link prediction task, obtaining RMSE of 0.92 and Hits@1 of 0.46. Details of the process and evaluation are given in Appendix I.

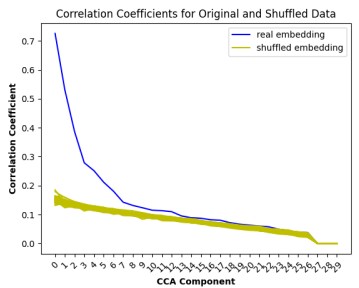
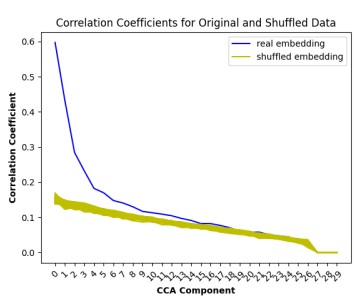
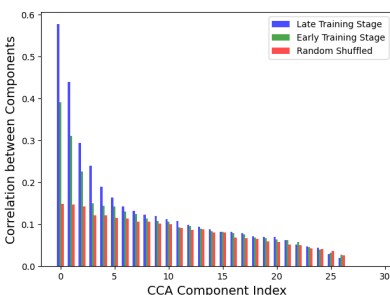

(a) Pearson's correlation coefficient (PCC) for true user-attribute pairings and 100 permuted pairings. Embeddings are computed by the additive scoring functions.

(b) Embeddings computed by multiplicative scoring function. Figure recreated from Guo et al. (2023)

(c) Correlation coefficients over different training stages of multiplicative scoring-function-based Knowledge Graph embedding

Figure 3: PCC is calculated between projected $\mathbf{A}$ and projected $\mathbf{U}$. $x$ axis stands for the $k$th components, $y$ axis gives the value.The PCC value for real pairings is larger than for any permuted pairings.

**Correlation-Based Linearity for KGEs** The continuous embedding matrix $\mathbf{U}_{corr} \in \mathbb{R}^{6040 \times 50}$ for correlation-based linearity detection experiment is populated with these embeddings. We also generate a binary attribute matrix $\mathbf{A}_{corr} \in \{0,1\}^{6040 \times 9}$ containing one 9-dimensional binary attribute vector for each user, including attributes based on gender, age, and occupation.

**Additive Compositionality for KGEs** For the additive compositionality experiment, we generate a binary attribute matrix $\mathbf{A}_{add} \in \{0,1\}^{14 \times 9}$ containing one 9-dimensional binary attribute vector for each user including attributes based on gender and age (2 genders $\times$ 7 age groups). The choice of attributes is made to match the choices in Guo et al. (2023). The continuous embedding matrix $\mathbf{U}_{add} \in \mathbb{R}^{14 \times 50}$ is generated by taking the mean across embeddings associated with the same attribute combination.

**Results** Results of the correlation-based compositionality detection method are reported in Figure 3a. Results from Guo et al. (2023) are recreated in Figure 3b for comparison. PCC, as measured by the correlation-based linearity detection method, is shown to be robust across these two different graph embedding methods. The additive scoring-function-based model (Figure 3a) shows a significant difference between real and random pairings, confirming that user embeddings encode demographic information.

Results of the additive compositionality detection experiment are reported in Figure 4, showing L2 loss of 0.04, cosine similarity of 0.97, and Hits@1 of 0.94, each outperforming the random baseline with $p = 0.01$. These findings reject the null hypothesis, demonstrating that user embeddings encode additive relationships between gender and age.

### 3.2.2 Linearity across Training Stages

We train embeddings for the MovieLens-1M dataset using the multiplicative scoring function (Yang et al. (2014)). Embeddings are extracted at both early (5 epochs) and late (300 epochs) stages of training. We assess the correlation-based linearity at each training stage. For each stage the attribute matrix $\mathbf{A}$ is the same as $\mathbf{A}_{corr}$ described above, whereas $\mathbf{U}_{early}$ and $\mathbf{U}_{late}$ contain embeddings from the early and late training stages respectively.

**Results** Results are reported in Figure 3c. We see that as training progresses, the correlation coefficient increases. This means that more demographic information becomes encoded into the embeddings.

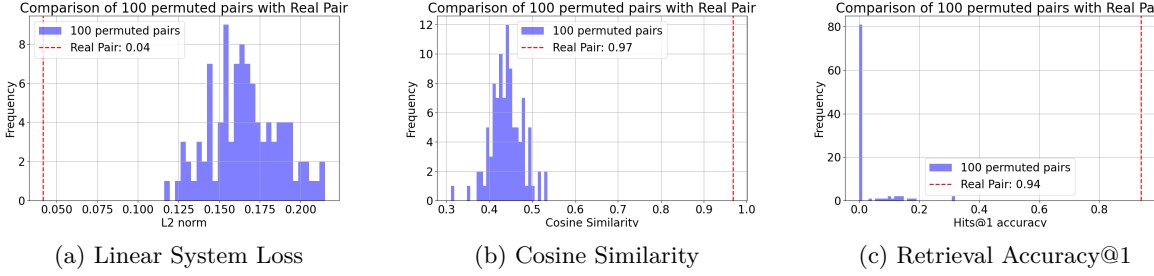

(a) Linear System Loss      (b) Cosine Similarity      (c) Retrieval Accuracy@1

Figure 4: The test statistics for user additive scoring function embedding decomposition. The dashed line represents the average performance of $\hat{\mathbf{U}}$ from user embeddings, while the bars show the distribution of results from 100 random permutations. (a) L2 loss, (b) Cosine Similarity, and (c) Retrieval Accuracy@5 compare real user embedding pairs to permuted pairs.

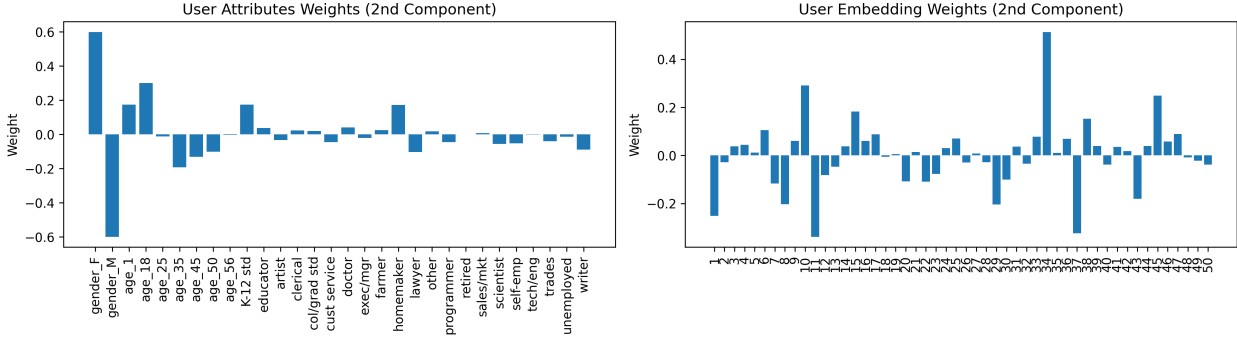

Figure 5: Positive weights in the second canonical component of CCA show the association between user attributes (e.g., *Female* and *Age: 18*) and user embedding dimensions (e.g., *34* and *10*).

### 3.2.3 Interpretation of Weights

Canonical Correlation Analysis (CCA) identifies linear combinations of variables, called canonical variables, from two datasets (user attributes and user embeddings) that are maximally correlated. The weights in CCA indicate the contribution of each original variable to the canonical variable, with the sign reflecting the direction of the relationship.

In Figure 5, positive weights on the user attribute side (e.g., *Female* and *Age: 18*) and user embedding side (e.g., dimensions *34* and *10*) highlight their strong contribution to their respective canonical variables. This strong correlation between user-side and movie-side canonical variables reveals meaningful links between demographic traits and movie preferences.

### 3.3 Word Embedding

We apply the same methods to examine two distinct signals contained in word2vec embeddings: semantic and syntactic information. To detect these signals, we use WordNet embeddings as semantic representation, and MorphoLex for syntactic structures. By comparing the word2vec embeddings against both WordNet and MorphoLex, we are able to disentangle the semantic and syntactic aspects of the word2vec representation.

### 3.3.1 Datasets

**WordNet Embeddings** WordNet (Miller, 1995) is a large lexical database of English that combines dictionary and thesaurus features with a graph structure. It organizes nouns, verbs, adjectives, and adverbs into synsets (sets of cognitive synonyms) interlinked by semantic and lexical relations. We use WN18RR (Dettmers et al., 2018), a subset of WordNet with 40,943 entities and 11 types of relation.

**MorphoLex** MorphoLex (Sánchez-Gutiérrez et al., 2018) provides a standardized morphological database derived from the English Lexicon Project, encompassing 68,624 words with nine variables for roots and affixes. An example extract from the dataset is given in Table 4. In this paper, we focus specifically on words with one root and multiple suffixes.

Table 4: Suffix presence (indicated by '1') for selected words from the MorphoLex dataset

| Word | er | t | y | est | ly | ness | less |
|------|----|----|----|-----|-----|------|------|
| weightier | 1 | 1 | 1 | 0 | 0 | 0 | 0 |
| weightiest | 0 | 1 | 1 | 1 | 0 | 0 | 0 |
| weightily | 0 | 1 | 1 | 0 | 1 | 0 | 0 |
| weightiness | 0 | 1 | 1 | 0 | 0 | 1 | 0 |
| weightlessly | 0 | 1 | 0 | 0 | 1 | 0 | 1 |

### 3.3.2 Experiments and Results

**Correlation-Based Linearity of Semantics and Morphology in Word2Vec** We examine the extent to which word2vec embeddings are correlated with WordNet embeddings and with MorphoLex representations. For WordNet, 25,781 words are selected from the intersection of the WN18RR and word2vec (GoogleNews-vectors-negative300) vocabularies. We train embeddings for these words over WordNet on an entity prediction task, which involves predicting the tail entity given a head entity and relation. Training details are provided in Appendix G.1. We build a continuous embedding matrix $\mathbf{U}_{w2v-wn} \in \mathbb{R}^{25,781 \times 300}$ using vectors from GoogleNews-vectors-negative300. We build a second continuous embedding matrix $\mathbf{U}_{wn-w2v} \in \mathbb{R}^{25,781 \times 20}$ and assess the correlation using the methods described in section 2.

To assess the correlation between word2vec and MorphoLex, 15,342 words and 81 suffixes are selected by intersecting the MorphoLex and word2vec (GoogleNews-vectors-negative300) vocabularies and filtering out suffixes occurring fewer than 10 times. We build a continuous embedding matrix $\mathbf{U}_{w2v-morpho} \in \mathbb{R}^{15,342 \times 300}$ using vectors from GoogleNews-vectors-negative300 and a binary embedding matrix $\mathbf{X}_{morpho} \in \{0,1\}^{15,342 \times 81}$. We again assess the correlation using the methods described in section 2.

As shown in Figures 6a and 6b, the Pearson Correlation Coefficient (PCC) between word2vec, WordNet embeddings, and MorphoLex binary vectors significantly exceeds the randomized baseline, indicating that word2vec embeddings, trained on contextual co-occurrences, implicitly capture both semantic and morphological information.

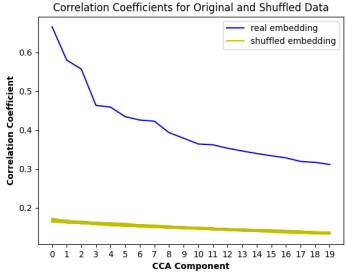

(a) PCC for the true WordNet-word2vec pairings and 50 permuted pairings.

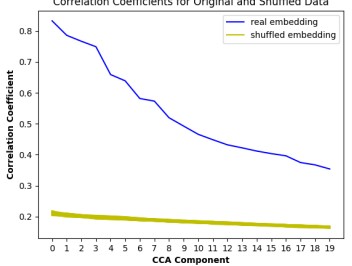

(b) PCC comparison for the true MorphoLex-word2vec pairings and 50 permuted pairings.

Figure 6: PCC is calculated between projected $\mathbf{A}$ and projected $\mathbf{U}$. The first 20 components are selected for illustration. Real pairings show higher PCC values than permuted ones, showing word2vec embeddings capture both semantic and morphological information.

**Additive Compositionality of word2vec Embeddings**   To examine the additive compositionality of word2vec embeddings across multiple suffixes, we select 278 words as follows. We filter words that have exactly 3 suffixes, each of which occurs 10 times or more[6]. This results in 17 suffixes, but some words end up with fewer than 3 suffixes because certain suffixes occur less than 10 times and are filtered out. We build a continuous embedding matrix $\mathbf{U}_{w2v-add} \in \mathbb{R}^{278 \times 300}$ using vectors from GoogleNews-vectors-negative300. We build a binary attribute matrix $\mathbf{A}_{morpho-add} \in \mathbb{R}^{278 \times 45}$ indicating the presence or absence of each root word and morphemes. We perform the additive compositionality experiment described in section 2 to determine whether embeddings may be decomposed into multiple suffixes. Results are presented in Figure 7. We see that word2vec embeddings can be decomposed into root and multiple suffixes fairly well. The linear system loss is 39.85, lower than the minimum loss of the random system (43.91). Cosine similarity is 0.44, greater than all instances of the random baseline, and retrieval accuracy @ 10 is greater than that of the random system. However, overall these values are low, showing that there is still a fair bit of information that is not being captured by this representation.

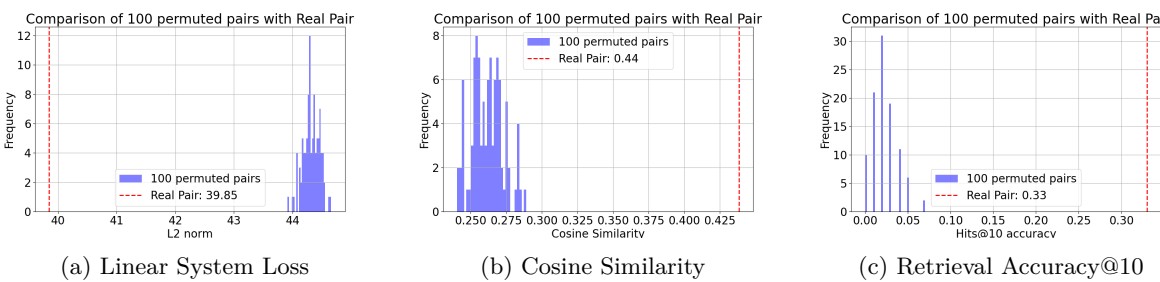

(a) Linear System Loss          (b) Cosine Similarity          (c) Retrieval Accuracy@10

Figure 7: The test statistics for word2vec embedding decomposition. Dash line is the average performance of $\hat{\mathbf{u}}$ learned from the word2vec embedding. The bars are the distribution of the results from random permutations that run for 100 times. (a) L2 loss, (b) Cosine Similarity, and (c) Retrieval Accuracy@10 compare real word2vec embedding pairs to permuted pairs.

## 4    Discussion and Conclusion

In many cases, embeddings are not easily explainable to humans, which may present safety concerns. To address this issue, we analyzed embeddings from word, sentence, and graph data structures and worked to interpret them in a more understandable way. Our central goal was to determine whether embeddings representing structured entities could be decomposed through additive composition. To this end, we applied two methods from the literature: (1) a correlation-based linearity analysis that measured how well learned embeddings correlated with known attributes, and (2) an additive compositionality detection method (Xu et al., 2023), which provides a way to decompose embeddings into vectors representing distinct attributes.

**Linearity, Decomposition and Compositionality**   Our paper adopted a two-step approach to explore the inherent structure of embeddings. In the first step, *Correlation-based linearity detection* confirmed a robust linear correlation between embeddings and candidate semantic features. This step established the necessary foundation for the second method, *additive compositionality detection,* where we rigorously demonstrated that final embeddings can be reconstructed as an additive combination of concept embeddings, using uniform coefficients analogous to those in word2vec's analogy tasks. Unlike the work in Arora et al. (2018), which focuses solely on decomposition, our *leave-one-out* experiment directly test the reconstruction of the original embedding, thereby providing a more stringent validation of additive compositionality. While previous research (Mitchell & Lapata, 2010; Coecke et al., 2010) has focused on constructing representations by combining parts into a whole, our approach inverted this task by decomposing holistic representations into their constituent components. This inversion allows us to examine compositionality from both the constructive and deconstructive perspectives. Our method builds on and extends earlier works (Maillard & Clark, 2015; Wijnholds et al., 2020; Baroni & Zamparelli, 2010; Lewis et al., 2022) by not only identifying these

---

[6]These words are listed in Appendix H

components but also validating their additive recombination via a *leave-one-out* strategy, which underscores the generative potential of the additive compositional structure in diverse embedding spaces.

The additive compositionality approach is fundamentally rooted in the linear system theory, as the superposition principle enables us to decompose embeddings into distinct component vectors that can then be recombined to reveal underlying relationships. In this way, linearity and additive compositionality are effectively the same: both describe how the properties of a whole can be derived from the sum of its parts.

Word2Vec (Mikolov et al., 2013b), for instance, set the goal for understanding compositionality by showing that vector arithmetic can capture meaningful relationships. This pioneering work has inspired further research on decomposing embeddings to discover compositional structures. We extended this line of inquiry by investigating compositionality across various data embeddings and evaluation metrics. Our *additive compositionality detection* method decomposes embeddings into constituent component vectors and then recombines them, clearly evidencing an additive signal that supports compositionality.

Our experiments demonstrated that embeddings across diverse domains exhibit additive compositionality, albeit to varying degrees. In word embeddings (e.g., word2vec), morphological information showed a weaker additive signal than expected. For sentence embeddings, we evaluated additive compositionality via decomposition into constituent concepts. Decomposing sentences into multiple concepts is inherently more complex due to interactions within token representations, making this a more stringent test of additive compositionality.

Different training techniques influence the degree of compositionality. In MultiBERTs, compositionality does not steadily increase with training and can decline over time (see Table 3). By contrast, compositionality in knowledge graph embeddings tends to rise steadily during training. Likewise, in SBERT, compositionality increases in earlier layers but declines slightly in later layers, presumably because these upper layers specialize in task-specific representations (Devlin et al., 2018; Tenney et al., 2019a). For graph embeddings, both additive and multiplicative scoring functions perform similarly, and further correlation-based analysis sheds light on the semantic meaning of embedding dimensions.

Our findings address the broader question of whether relations between structured entities can be captured through simple vector operations. Evaluating a range of word, sentence, and graph embeddings showed a common thread of additive compositionality. For sentence embeddings, conceptual components can often be combined additively, indicating that even large, non-linear models (e.g., GPT or Llama) retain interpretable vector structures. Word embeddings effectively encode both morphological and semantic relationships, as illustrated by transformations like *weight + y + ly = weightily*. In knowledge graph embeddings, attributes such as user demographics (age, gender) can be adjusted via vector operations, enabling inference of new relationships in recommender systems.

Overall, these results underscore that embeddings from diverse sources maintain a surprising degree of additive compositional structure, which can be leveraged to enhance interpretability in representation learning. Future work will build on this foundation by exploring debiasing techniques (Bose & Hamilton, 2019) and examining how large language models produce and utilize additive components in tasks like reasoning. Additionally, given the large size of LLaMA embeddings, we plan to explore dimensionality reduction techniques, such as those inspired by the Johnson–Lindenstrauss lemma, to preserve statistical properties while improving numerical stability and computational efficiency. A more detailed investigation at both the layer and attention-head level, as well as an extension to additional graph embedding models, may further illuminate how different architectures preserve and exploit these additive relationships.

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

# A    Details of Correlation-Based Compositionality Detection Method

Canonical Correlation Analysis (CCA) is used to measure the correlation information between two multivariate random variables (Shawe-Taylor et al., 2004). Just like the univariate correlation coefficient, it is estimated on the basis of two aligned samples of observations.

A matrix of binary-valued attribute embeddings, denoted as $\mathbf{A}$, is essentially a matrix representation where each row corresponds to a specific attribute and each column corresponds to an individual data point (such as a word, image, or user). The entries of the matrix can take only two values, typically 0 or 1, signifying the absence or presence of a particular attribute. For example, in the context of textual data, an attribute might represent whether a word is a noun or not, and the matrix would be populated with 1s (presence) and 0s (absence) accordingly.

On the other hand, a matrix of user embeddings, denoted as $\mathbf{U}$, is a matrix where each row represents an individual user, and each column represents a certain feature or dimension of the embedding space. These embeddings are continuous-valued vectors that capture the movie preference of the users. The values in this matrix are not constrained to binary values and can span a continuous range.

These paired random variables are often different descriptions of the same object, for example genetic and clinical information about a set of patients (Seoane et al., 2014), French and English translations of the same

document (Vinokourov et al., 2002), and even two images of the same object from different angles (Guo & Wu, 2019).

In the example of viewers and movies, we use this method to compare two descriptions of users. One matrix is based on demographic information, which are indicated by Boolean vectors. The other matrix is based on their behaviour, which is computed by their movie ratings only.

Assuming we have a vector for an individual user's attribute embedding, denoted as:

$$\mathbf{a} = (a_1, a_2, \ldots, a_n)^T$$

and a corresponding individual user's computed embedding:

$$\mathbf{u} = (u_1, u_2, \ldots, u_m)^T$$

we aim to explore the correlation between these two representations across multiple users. Given a set of $q$ users, we define $\mathbf{A}$ as a $q \times n$ matrix where each row corresponds to the attribute embeddings for a specific user, and $\mathbf{U}$ as a $q \times m$ matrix where each row represents the computed embedding of the same user. Here, $n$ is the number of attributes, and $m$ is the dimensionality of the user embeddings.

Canonical Correlation Analysis (CCA) is then employed to find the projection matrices $\mathbf{W}_A \in \mathbb{R}^{n \times k}$ and $\mathbf{W}_U \in \mathbb{R}^{m \times k}$ that maximise the correlation between the transformed representations of $\mathbf{A}$ and $\mathbf{U}$. Each projection matrix contains $k$-projection vectors, where $k$ is the number of canonical components that depend on the eigenvalues of the covariance matrix. For each $k$-th canonical component, the projections of the attribute and user embeddings are given by:

$$\mathbf{A}_k = \mathbf{A}\mathbf{w}_{a_k} \quad \text{and} \quad \mathbf{U}_k = \mathbf{U}\mathbf{w}_{u_k}$$

where $\mathbf{w}_{a_k} \in \mathbb{R}^n$ and $\mathbf{w}_{u_k} \in \mathbb{R}^m$ are the $k$-th projection vectors from the matrices $\mathbf{W}_A$ and $\mathbf{W}_U$, respectively. $\mathbf{A}_k \in \mathbb{R}^q$ represents the projection of the original attribute matrix $\mathbf{A}$ (size $q \times n$) onto the $k$-th canonical direction, using the projection vector $\mathbf{w}_{a_k} \in \mathbb{R}^n$. It results in a vector of size $q \times 1$ that contains the transformed values for each user for the $k$-th component. Similarly, $\mathbf{U}_k \in \mathbb{R}^q$ represents the projection of the original user embedding matrix $\mathbf{U}$ (size $q \times m$) onto the $k$-th canonical direction, using the projection vector $\mathbf{w}_{u_k} \in \mathbb{R}^m$. It also results in a vector of size $q \times 1$ that contains the transformed values for each user for the $k$-th component.

The goal of CCA is to maximise the Pearson Correlation Coefficient (PCC) between these transformed representations, i.e., between $\mathbf{A}_k$ and $\mathbf{U}_k$, for each $k$-th canonical component. The correlation for the $k$-th canonical component, denoted as $\rho_k$, is given by the formula:

$$\rho_k = \frac{\sum_{i=1}^q \left((\mathbf{A}_k)_i - \mu_{\mathbf{A}_k}\right)\left((\mathbf{U}_k)_i - \mu_{\mathbf{U}_k}\right)}{\sqrt{\sum_{i=1}^q \left((\mathbf{A}_k)_i - \mu_{\mathbf{A}_k}\right)^2}\sqrt{\sum_{i=1}^q \left((\mathbf{U}_k)_i - \mu_{\mathbf{U}_k}\right)^2}} \tag{1}$$

where $\mu_{\mathbf{A}_k}$ and $\mu_{\mathbf{U}_k}$ are the means of the transformed attribute and user embeddings, respectively, and $q$ is the number of users.

In matrix form, we can express this objective as maximising the correlation between the transformed matrices $\mathbf{A}\mathbf{W}_A$ and $\mathbf{U}\mathbf{W}_U$. This is formalised as:

$$\rho = \max_{\mathbf{W}_A, \mathbf{W}_U} \text{corr}\left(\mathbf{A}\mathbf{W}_A, \mathbf{U}\mathbf{W}_U\right) \tag{2}$$

where the correlation is maximised across the projection matrices $\mathbf{W}_A$ and $\mathbf{W}_U$, and the result is a set of $K$-canonical correlations $\rho_1, \rho_2, \ldots, \rho_k$ that describe the relationship between the attribute embeddings and the user embeddings for the entire dataset.

Thus, by computing the canonical correlations $\rho_K$ for each component, we obtain insights into how well the attribute embeddings and computed user embeddings are aligned in terms of their underlying structure.

As shown in Figure 8, consider the case where we have 4 users. The attribute matrix $\mathbf{A}$ has dimensions $4 \times 4$, representing 4 users and 4 attributes, and the user embedding matrix $\mathbf{U}$ has dimensions $4 \times 3$, representing 4

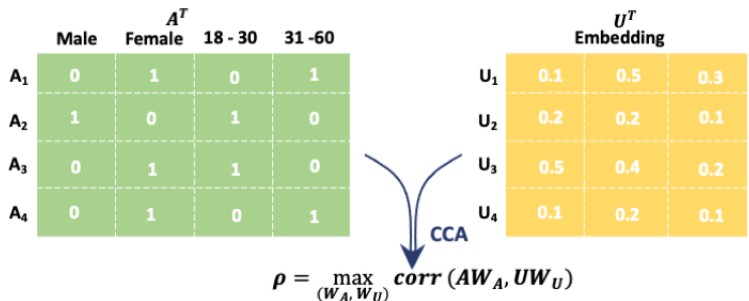

Figure 8: Schematic of Correlation-based compositionality Detection (Guo et al., 2023)

users and 3 embedding dimensions. Thus: - $\mathbf{A} \in \mathbb{R}^{4 \times 4}$: the attribute matrix where each row represents the 4-dimensional attribute vector for a user. - $\mathbf{U} \in \mathbb{R}^{4 \times 3}$: the user embedding matrix where each row represents the 3-dimensional embedding vector for a user.

Canonical Correlation Analysis (CCA) is employed to find the projection matrices $\mathbf{W}_A \in \mathbb{R}^{4 \times k}$ and $\mathbf{W}_U \in \mathbb{R}^{3 \times k}$ that maximize the correlation between the transformed representations of $\mathbf{A}$ and $\mathbf{U}$. Each projection matrix contains $k$-projection vectors, where $k$ is the number of canonical components that depend on the eigenvalues of the covariance matrix. For each $k$-th canonical component, the projections of the attribute and user embeddings are given by:

$$\mathbf{A}_k = \mathbf{A}\mathbf{w}_{a_k} \quad \text{and} \quad \mathbf{U}_k = \mathbf{U}\mathbf{w}_{u_k}$$

where $\mathbf{w}_{a_k} \in \mathbb{R}^4$ and $\mathbf{w}_{u_k} \in \mathbb{R}^3$ are the $k$-th projection vectors from the matrices $\mathbf{W}_A$ and $\mathbf{W}_U$, respectively.

# B    Leave-one-out experiment

1. **Leave Out**: Exclude the $i$-th row from $\mathbf{A}$ and $\mathbf{U}$ to obtain $\mathbf{A}_{-i}$ and $\mathbf{U}_{-i}$.

2. **Train**: Solve $\mathbf{A}_{-i}\mathbf{X} = \mathbf{U}_{-i}$ using the pseudo-inverse to obtain $\mathbf{X}$.

3. **Predict**: Estimate the left-out embedding using $\hat{\mathbf{u}}_i = \mathbf{a}_i\mathbf{X}$, where $\mathbf{a}_i$ is the attribute vector of entity $i$.

4. **Evaluate**: Compare $\hat{\mathbf{u}}_i$ with the actual embedding $\mathbf{u}_i$ using the following metrics:

   (a) *L2 Loss*: $L_2 = \|\hat{\mathbf{u}}_i - \mathbf{u}_i\|^2$, measuring the reconstruction error.

   (b) *Embedding Prediction (Cosine Similarity)*: $\cos(\theta) = \dfrac{\hat{\mathbf{u}}_i \cdot \mathbf{u}_i}{\|\hat{\mathbf{u}}_i\| \|\mathbf{u}_i\|}$, assessing the alignment between the predicted and actual embeddings.

   (c) *Identity Prediction (Retrieval Accuracy)*: Determines if $\hat{\mathbf{u}}_i$ correctly identifies entity $i$ by checking if $\mathbf{u}_i$ is the nearest neighbor to $\hat{\mathbf{u}}_i$ among all embeddings.

# C    Knowledge Graph Embedding

**Knowledge Graph Embedding**    A *graph* $G = (V, E)$ consists of a set of vertices $V$ with edges $E$ between pairs of vertices. In a *knowledge graph*, the vertices $V$ represent entities in the real world, and the edges $E$ encode that some relation holds between a pair of vertices. As a running example, we consider the case where the vertices $V$ are a set of viewers and films, and the edges $E$ encode the fact that a viewer has rated a film.

Knowledge Graphs represent information in terms of entities (or nodes) and the relationships (or edges) between them. The specific relation $r$ that exists between two entities is depicted as a directed edge, and this connection is represented by a triple $(h, r, t)$. In this structure, we distinguish between the two nodes

involved: the *head* ($h$) and the *tail* ($t$), represented by vectors $\mathbf{h}$ and $\mathbf{t}$ respectively. Such a triple is termed a *fact*, denoted by $f$:

$$f = (h, r, t)$$

**Embedding Function.** A knowledge graph $G = (V, E)$ can be embedded by assigning each node $v \in V$ a vector $\mathbf{x}_v \in \mathbb{R}^d$. This embedding function, $\Phi_{KG} : V \to \mathbb{R}^d$, maps nodes into a continuous space where their relationships are captured by a scoring or distance function. For instance, one could define a threshold $\theta$ such that an edge $(v_i, v_j) \in E$ exists if and only if $D(\mathbf{x}_{v_i}, \mathbf{x}_{v_j}) < \theta$. Conversely, given a set of embedded points, links between nodes can be recovered by applying a learned scoring function $S(\mathbf{x}_{v_i}, \mathbf{x}_{v_j})$.

## C.1 Multiplicative Scoring

Nickel et al. (2011) introduced a tensor-factorization approach for relational learning, treating each frontal slice of a three-dimensional tensor as a co-occurrence matrix for a given relation. In this model, a triple $(h, r, t)$ is scored using embeddings $\mathbf{h}, \mathbf{R}, \mathbf{t}$, where $\mathbf{h}, \mathbf{t} \in \mathbb{R}^d$ represent head and tail entities, and $\mathbf{R} \in \mathbb{R}^{d \times d}$ represents the relation:

$$S(h, r, t) = \mathbf{h}^T \mathbf{R} \, \mathbf{t}. \tag{3}$$

Various specializations exist, such as DistMult (Yang et al., 2014), which restricts $\mathbf{R}$ to a diagonal matrix (reducing overfitting), and ComplEx (Trouillon et al., 2016), which employs complex-valued embeddings for asymmetric relations. In this work, we adopt DistMult due to its simplicity and scalability, particularly its suitability for large knowledge graphs.

## C.2 Additive Scoring

TransE (Bordes et al., 2013) introduces a translation-based perspective, where each relation is a vector that shifts the embedding of the head entity to the tail entity. A triple $(h, r, t)$ is scored by:

$$S(h, r, t) = \|\mathbf{h} + \mathbf{r} - \mathbf{t}\|, \quad \mathbf{h}, \mathbf{r}, \mathbf{t} \in \mathbb{R}^d. \tag{4}$$

For example, *King + FemaleOf ≈ Queen*. This translation idea captures relational semantics by minimizing the distance between $\mathbf{h} + \mathbf{r}$ and $\mathbf{t}$.

**Rating Prediction** In alignment with (Berg et al., 2017), we establish a function $P$ that, given a triple of embeddings $(\mathbf{h}, \mathbf{R}, \mathbf{t})$, calculates the probability of the relation against all potential alternatives.

$$P(\mathbf{h}, \mathbf{R}, \mathbf{t}) = \text{SoftArgmax}(S(f)) = \frac{e^{S(f)}}{e^{S(f)} + \sum_{r' \neq r \in \mathscr{R}} e^{S(f')}} \tag{5}$$

In the above formula, $f = (h, r, t)$ denotes a true triple, and $f' = (h, r', t)$ denotes a corrupted triple, that is a randomly generated one, that we use as a proxy for a negative example (a pair of nodes that are not connected).

Assigning numerical values to relations $r$, the predicted relation is then just the expected value prediction $= \sum_{r \in \mathscr{R}} r P(\mathbf{h}, \mathbf{R}, \mathbf{t})$ In our application of viewers and movies, the set of relations $\mathscr{R}$ could be the possible ratings that a user can give a movie. The predicted rating is then the expected value of the ratings, given the probability distribution produced by the scoring function. $S(f)$ refers to the scoring function in Yang et al. (2014).

To learn a graph embedding, we follow the setting of Bose & Hamilton (2019) as follows,

$$L = -\sum_{f \in \mathscr{F}} \log \frac{e^{S(f)}}{e^{S(f)} + \sum_{f' \in \mathscr{F}'} e^{S(f')}} \tag{6}$$

This loss function maximizes the probabilities of true triples ($f$) and minimizes the probability of triples with corrupted triples: ($f'$).

**Evaluation Metrics**   We use 4 metrics to evaluate our performance on the link prediction task. These are root mean square error (RMSE, $\sqrt{\frac{1}{n}\sum_{i=1}^{n}(\hat{y}_i - y_i)^2}$, where $\hat{y}_i$ is our predicted relation and $y_i$ is the true relation), Hits@K - the probability that our target value is in the top $K$ predictions, mean rank (MR) - the average ranking of each prediction, and mean reciprocal rank (MRR) to evaluate our performance on the link prediction task. These are standard metrics in the knowledge graph embedding community.

## D   Additive Compositionality by Model

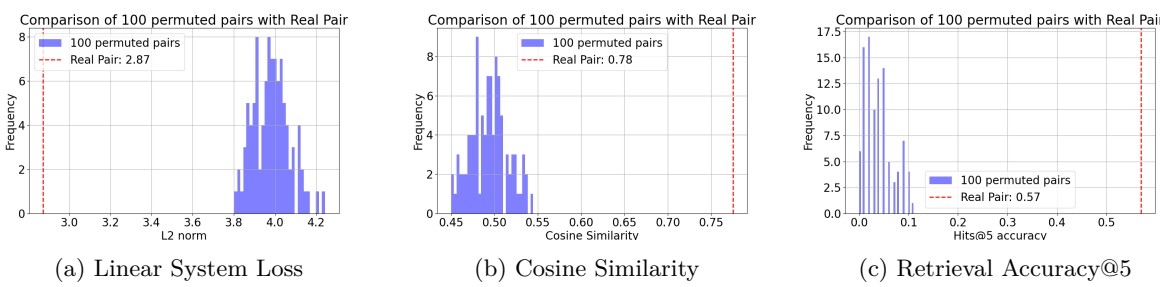

(a) Linear System Loss             (b) Cosine Similarity             (c) Retrieval Accuracy@5

Figure 9: Test statistics for GPT embedding decomposition. Dashed line is the average performance of $\hat{\mathbf{U}}$ learned from the user embedding. Bars are the distribution of the results from 100 random permutations.

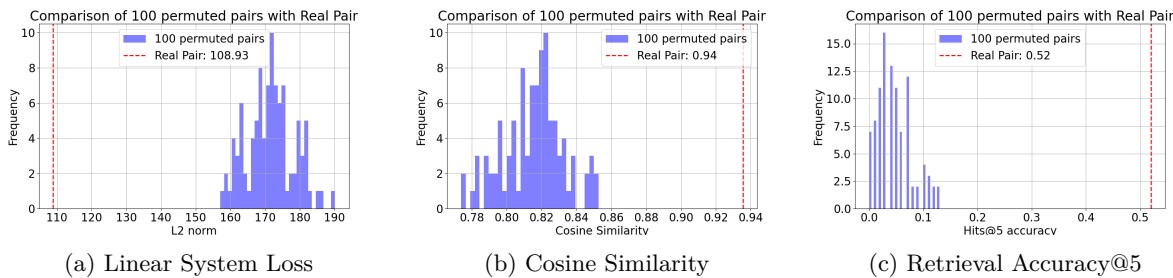

(a) Linear System Loss             (b) Cosine Similarity             (c) Retrieval Accuracy@5

Figure 10: Test statistics for Llama embedding decomposition. Dashed line is the average performance of $\hat{\mathbf{U}}$ learned from the user embedding. Bars are the distribution of the results from 100 random permutations.

## E   Compositionality across Layers and Training Stages

### E.1   Comparison of Different Layers

**Comparison Metrics**   To fairly compare different layers, we cannot rely solely on raw cosine similarities or retrieval accuracies due to variations in scales and distributions. Instead, we use normalized metrics for comparability. The **Normalized Cosine Similarity** computes the difference between the mean real similarity and the mean permuted similarity, normalized by the maximum possible difference, $(1 - \text{Mean Permuted Similarity})$. The **Absolute Difference** is a simple measure of the difference between mean real and permuted similarities. Lastly, the **Relative Difference (Percentage Improvement)** expresses this difference as a percentage of the permuted similarity, indicating proportional improvement. These metrics enable robust and fair comparisons across models.

Table 5: Additive Compositionality Metrics Across SBERT Layers

| Layer | Mean Sim (Real) | Mean Sim (Permuted) | Norm. Cosine Sim | Hits@5 Acc (Real) | Hits@5 Acc (Permuted) | Norm. Retrieval Acc |
|---|---|---|---|---|---|---|
| 0 | 0.8889 | 0.7808 | 0.4930 | 0.59 | 0.0397 | 0.5731 |
| 1 | 0.9366 | 0.8671 | 0.5228 | 0.57 | 0.0406 | 0.5518 |
| 2 | 0.9397 | 0.8576 | 0.5767 | 0.61 | 0.0390 | 0.5942 |
| 3 | 0.9403 | 0.8330 | 0.6424 | 0.64 | 0.0412 | 0.6245 |
| 4 | 0.9408 | 0.8298 | 0.6523 | 0.66 | 0.0400 | 0.6458 |
| 5 | 0.9409 | 0.8273 | 0.6577 | 0.62 | 0.0417 | 0.6035 |
| 6 | 0.7761 | 0.4865 | 0.5640 | 0.59 | 0.0405 | 0.5727 |

Table 6: Additive Compositionality Metrics at Different Training Steps of BERT

| Model | Training Steps | Mean Sim (Real) | Mean Sim (Permuted) | Norm. Cosine Sim | Hits@5 Acc (Real) | Permuted Acc | Norm. Retrieval Acc |
|---|---|---|---|---|---|---|---|
| cls_0k | 0 | 0.9884 | 0.9882 | 0.0163 | 0.44 | 0.0418 | 0.4156 |
| cls_20k | 20,000 | 0.8787 | 0.7722 | 0.4676 | 0.55 | 0.0407 | 0.5309 |
| cls_40k | 40,000 | 0.8773 | 0.7724 | 0.4607 | 0.48 | 0.0405 | 0.4581 |
| cls_100k | 100,000 | 0.9201 | 0.8323 | 0.5236 | 0.55 | 0.0417 | 0.5304 |
| cls_1000k | 1,000,000 | 0.9545 | 0.9149 | 0.4655 | 0.44 | 0.0408 | 0.4162 |
| cls_2000k | 2,000,000 | 0.9538 | 0.9094 | 0.4896 | 0.48 | 0.0415 | 0.4575 |

# F  Error Bar Plot

# G  Word Embeddings

## G.1  WordNet Embedding

We want to ensure our WordNet embedding can contain the semantic relation in it. Therefore, we train the embedding with the task of predicting the tail entity given a head entity and relation. For example, we might want to predict the hypernym of cat:

$$< \textbf{cat}, hypernym, \textbf{?} >$$

**Mapping Freebase ID to text**  WordNet is constructed with Freebase ID only, an example triple could be <00260881, hypernym, 00260622>. We follow villmow (2019) to preprocess the data and map each entity with the text with a real meaning.

The above triple can then be processed with the real semantic meaning: <land reform, hypernym, reform>. The word2vec word embedding is pretrained from a google news corpus. We train the WordNet Embedding in the following way:

1. We split our dataset to use 90% for training, 10% for testing.

2. Triples of $(head, relation, tail)$ are encoded as relational triples $(h, r, t)$.

3. We randomly initialize embeddings for each $h_i$, $r_j$, $t_k$, use the scoring function in Equation 4 and minimize the loss by Margin Loss.

4. We sampled 20 corrupted entities. Learning rate is set at 0.05 and training epoch at 300.

Results can be found in the Table 7, which shows that our WordNet embeddings do contain semantic information.

Table 7: Link prediction performance for WordNet

|  | Hits@1 | Hits@3 | Hits@10 | MRR |
|---|---|---|---|---|
| WordNet | 0.39 | 0.41 | 0.43 | 0.40 |

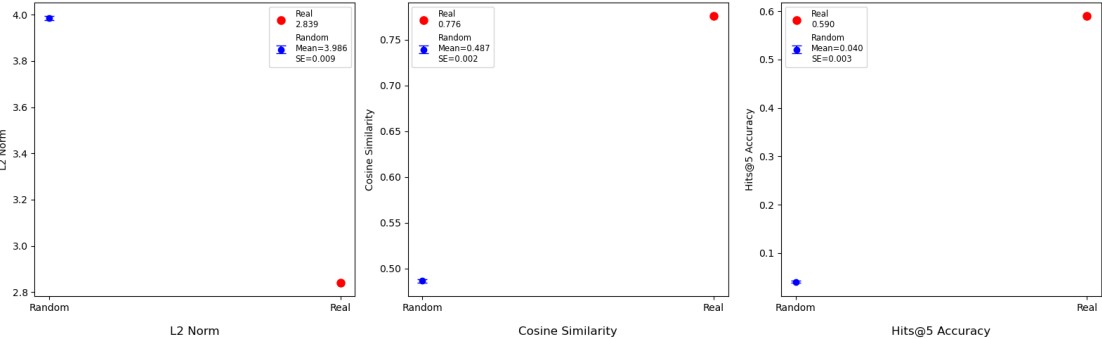

Figure 11: Error bar plot comparing the real measured values (red dots) with the corresponding random permutation distributions (blue markers with error bars) for three metrics: (a) L2 norm, (b) cosine similarity, and (c) Hits@5 accuracy. The blue error bars represent the mean ± standard error of the metric computed over 100 random permutations, thereby quantifying the variability and uncertainty in the random baseline.

## H  List of words of experiments: Decomposing word2vec Embedding by Additive Compositionality Detection

allegorically, whimsicality, whimsically, voyeuristically, weightier, weightiest, weightily, weightiness, weightlessly, veritably, visualizations, tyrannically, traitorously, transcendentally, transitionally, tangentially, temperamentally, surgically, structurally, studiously, studiousness, stylistically, spiritualistic, slipperiest, slipperiness, serviceability, serviceably, sectionalism, sentimentalism, sentimentalist, sentimentalized, sentimentalizes, sentimentalizing, sentimentally, serialization, serializations, satanically, reverentially, ritualistically, regularization, quizzically, rapturously, puritanically, probationers, probationer, psychiatrically, preferentially, practicably, practicalities, pleasurably, polarizations, phenomenally, personalizations, pessimistically, pathetically, occupationally, optionally, oratorically, nationalizations, nautically, neutralizer, neutralizers, mysteriously, mysteriousness, narcissistically, moralistically, melodiously, melodiousness, memorializes, memorializing, metrication, materialistically, mechanistically, mechanizations, longitudinally, lexically, liberalization, liquidator, liquidators, journalistic, inferentially, injuriously, hysterically, idealistically, heretically, futuristically, fractionally, fluoridation, fictionalized, fictionalizes, fictionalizing, figuratively, farcically, fatalistic, environmentalists, environmentally, episodically, equitably, emotionlessly, ecclesiastically, editorialized, editorializes, editorializing, editorially, educationalist, educationalists, educationally, egotistical, egotistically, dictatorships, differentiations, derivatively, developmentally, deviationist, deviationists, decoratively, deferentially, definitively, demagogically, demonically, decimalization, cumulatively, conversationalists, conversationally, confidentialities, conspiratorially, collectivization, colonialists, classically, chauvinistically, censoriousness, certifications, capitalizations, breathalyser, breathalysers, brutalization, antagonistically, apocalyptically, weightlessness, westernization, victoriously, visualization, vocalization, urbanization, Unitarian, Unitarians, transcendentalism, transcendentalist, transcendentalists, theatrically, theoretically, technicalities, technicality, technically, speculatively, socialistic, socialization, sophisticate, sophisticated, sophisticates, significantly, sensational, sentimentalists, sentimentality, sentimentalize, scientifically, satirically, rotationally, residentially, relativistic, realistically, prudentially, pressurization, probabilistic, probationary, professionalism, professionally, popularization, potentialities, potentiality, potentially, practicability, practicality, practically, polarization, phosphorescence, phosphorescent, physically, physicalness, periodically, personalization, particularistic, paternalistic, operational, oratorical, organizational, normalization, numerically, negatively, negativism, neutralization, nationalistic, nationalization, naturalistic, naturalization, mechanically, memorialize, memorialized, metrically, maturational, localization, linguistically, liquidation, liquidations, juridical, justifiably, industrialization, imperialistic, incidentally, identically, imaginatively, historically, harmoniously, graphically, generalization, generalizations, fictionalize, existentialism, existentialist, existentialists, evangelicalism, environmentalism, environmentalist, equalization, equalizers, equatorial, electrically, electronically, emotionalism, emotionality, emotionally, economically, editorialist, editorialize, directionality, directionally, dictatorial, dictatorship, differentiation, conversationalist, confidentiality, confidentially, colonialism, colo-

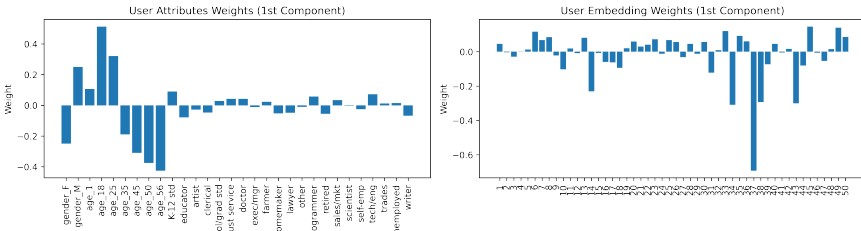

Figure 12: Positive weights in the first canonical component of CCA show the association between user attributes (e.g., *male* and *Age: 18*) and user embedding dimensions (e.g., *44* and *49*).

nialist, commercialization, communicational, civilizational, centralization, certification, chemically, capitalistic, capitalization, catastrophically, categorically, behaviorally, authentication, authentications, authenticator, artistically, architecturally, anatomically, Anglicanism, alternatively, altruistically, adventurously, acoustically, activation, additionally

## I Movie-Lens Training Details

This experiment was conducted on the MovieLens 1M dataset (Harper & Konstan, 2015) which consists of a large set of movies and users, and a set of movie ratings for each individual user. It is widely used to create and test recommender systems. Typically, the goal of a recommender system is to predict the rating of an unrated movie for a given user, based on the rest of the data. The dataset contains 6040 users and approximately 3900 movies. Each user-movie rating can take values in 1 to 5. There are 1 million triples (out of a possible $6040 \times 3900 = 23.6m$), so that the vast majority of user-movie pairs are not rated.

Users and movies each have additional attributes attached. For example, users have demographic information such as gender, age, or occupation. Whilst this information is typically used to improve the accuracy of recommendations, we use it to test whether the embedding of a user correlates to private attributes, such as gender or age. We compute our graph embedding based only on ratings, leaving user attributes out. Experiments for training knowledge graph embeddings are implemented with the OpenKE (Han et al., 2018) toolkit. We train our model on GeForce GTX TITAN X.

We embed the knowledge graph in the following way:

1. We split our dataset to use 90% for training, 10% for testing.

2. Triples of $(user, rating, movie)$ are encoded as relational triples $(h, r, t)$.

3. We randomly initialize embeddings for each $h_i$, $r_j$, $t_k$ and train embeddings to minimize the loss in equation 6 [7].

4. We sampled 10 corrupted entities and 4 corrupted relations per true triple. Learning rate is set at 0.01 and training epoch at 300.

We verify the quality of the embeddings by carrying out a link prediction task on the remaining 10% test set. We achieved a RMSE score of 0.88, Hits@1 score of 0.46 and Hits@3 as 0.92, MRR as 0.68 and MR as 1.89.

## J Analysis of the First Canonical Component of CCA for Knowledge Graph Embedding

---

[7] We add a negative sign for the additive scoring function, since we want to maximize the probability of the true triple, which aligns the setting of this loss function

