# OpenReview forum: "Understanding Compositionality in Data Embeddings"
_TMLR — Rejected by TMLR_

### Review · Reviewer_GqpX · 2025-02-17

**Summary Of Contributions:**

The paper explores (mainly textual) high-dimensional embeddings, verifying whether they correlate with the true known attributes of the considered entity (e.g., a sentence), and if they show "additive compositionality", i.e. if an entity (in embedding space) can be represented as a linear combination of the embeddings of the attributes.

They apply 2 methods to verify the two properties above, and they investigate word, sentence, and graph embeddings. Furthermore, the Authors look at how depth and training time affect their findings.

This paper is purely empirical, with claim of providing original contribution in showing the extend of sematic representation given by the embeddings.

**Audience:**

Yes

**Claims And Evidence:**

Yes

**Requested Changes:**

An important adjustement for my recommendation is a rewriting of the initial sections. This follows the mentioned 3 main weaknesses. The authors could be more schematic in the initial presentation of their method, their contribution, their findings, and the related work. This would involve a slightly more explicit definition of the measured variables in the paper (like, e.g., $\rho$), to avoid vagueness in the initial sections. Furthermore, a more clear and formal definition of the maximization in the first equation is, in my opinion, crucial.

Furthermore, the Authors could consider rephrasing their contribution more in terms of the quantitative explorative findings they have, and less on the effective statistical significance of embeddings, which is rather reasonable and maybe less surprising / interesting.

**Strengths And Weaknesses:**

**Strengths**

1 - The Authors take an effort in investigating a large number of different datasets, to explore both word and sentence embeddings, as well as graph ones.

2 - Exhaustive investigation of high dimensional embeddings is an important research direction. Besides providing better intuition on the foundational aspects and interpretability of, e.g., large language models; it can be helpful to verify the power and limitation of these structures for different learning tasks.

**Weaknesses**

1 - I did not find the initial part of the paper exceptionally readable. The *Motivation* paragraph looks more like an outline of the work, and the *Findings* paragraph is also not very clear about what are the actual findings, sticking to rather vague statements as "...indicating that even large, non-linear models preserve interpretable structures...". To be fair, it is hard at this point to give an exact formulation of the findings, which depends on Section 2.2, but in the current state it is a bit confusing.

2 - Section 1.1 resembles a *Related Work* section, however with included comments (at the end of the paragraphs) on how this work tackles the related topic, and how then it differs from previous work. Maybe, to improve readability, I would suggest making clear what the contribution of this paper is from the introduction, and limiting this other section as a place to provide description / context / bibliography for the separate topics. This comment also addresses Section 2.1, which again merges together what done in previous work and the particular approach followed by this paper.

3 - Section 2.2 introduces $\rho$, defining it via a maximization of a multidimensional object (I believe the correlation is empirically measured averaging on the different entities, while this is also not clearly explained), which is not clearly defined. How is $k$ chosen here? A projection matrix is generally defined such that $P^2 = P$, but here we are dealing with rectangular matrices. While it is qualitatively clear what this quantity represents, the Authors should really formally introduce all the settings required to fully define $\rho$, what a CAA component is, and examples would be helpful. Always in this direction, the estimation of $\rho$ could probably be meaningful only when $q$ is *quite* larger than the other scalings, which might be a relevant problem when investigating Llama embeddings, see the second minor comment below.

**Minor comments**

LOO (for leave-one-out) is an introduced but never used shorthand.

Why are Llama embeddings too large in the footnote 5? Is this a numerical issue? The Author could consider projecting the embeddings in a lower dimensional space to preserve the statistical properties of the embeddings while benefitting numerics (https://en.wikipedia.org/wiki/Johnson%E2%80%93Lindenstrauss_lemma)

The baseline the authors look at show some degree of alignement between random embeddings (e.g. Figure 1, yellow line is significantly larger than 0). This is commented right before Table 2, as a consequence of a specific semantic space all sentences belong to. Similar qualitative behaviour is seen also for word and graph embeddings. I would avoid the comment on the semantic space, unless empirical validation is provided, such as showing close to 0 when the dataset contains semantically very different sentences.

In Figure 5, why do the authors consider the second canonical component of CCA? This could be suspicious, why was this preferred on the 1st component?

---

> ### Author Response · Authors · 2025-03-14
> **Response to reviewer GqpX**
>
> Thank you for your thorough review; your insights and suggestions are greatly appreciated and have been extremely helpful. We have uploaded a revision to address the requested changes.
> > An important adjustment for my recommendation is a rewriting of the initial sections. This follows the mentioned 3 main weaknesses. The authors could be more schematic in the initial presentation of their method, their contribution, their findings, and the related work.
>
> Thank you for your insightful review and this is very helpful. We have revised the initial sections to clarify the introduction and motivation (page 1), contributions (page 2), and related work (Section 1.1).
>
> > This would involve a slightly more explicit definition of the measured variables in the paper (like, e.g., ρ), to avoid vagueness in the initial sections. Furthermore, a more clear and formal definition of the maximization in the first equation is, in my opinion, crucial.
>
> Below we provide a full definition of \rho, and we put it in the appendix A (page 17) of the revised paper. We have also provided an example in Appendix A.
>
> Assuming we have a vector for an individual user's attribute embedding, denoted as:
> \\[
> \\mathbf{a} = \\left( a_1, a_2, \\dots, a_n \\right)^T
> \\]
> and a corresponding individual user's computed embedding:
> \\[
> \\mathbf{u} = \\left( u_1, u_2, \\dots, u_m \\right)^T
> \\]
> we aim to explore the correlation between these two representations across multiple users. Given a set of \\( q \\) users, we define \\( \\mathbf{A} \\) as a \\( q \\times n \\) matrix where each row corresponds to the attribute embeddings for a specific user, and \\( \\mathbf{U} \\) as a \\( q \\times m \\) matrix where each row represents the computed embedding of the same user. Here, \\( n \\) is the number of attributes, and \\( m \\) is the dimensionality of the user embeddings.
>
> Canonical Correlation Analysis (CCA) is then employed to find the transformation matrices \\( \\mathbf{W}\_A \\in \\mathbb{R}^{n \\times k} \\) and \\( \\mathbf{W}\_U \\in \\mathbb{R}^{m \\times k} \\) that maximise the correlation between the transformed representations of \\( \\mathbf{A} \\) and \\( \\mathbf{U} \\). Each transformation matrix contains \\( k \\)-transformation vectors, where \\( k \\) is the number of canonical components that depend on the eigenvalues of the covariance matrix. For each \\( k \\)-th canonical component, the transformations of the attribute and user embeddings are given by:
> \\[
> \\mathbf{A}\_k = \\mathbf{A} \\mathbf{w}\_{a_k} \\quad \\text{and} \\quad \\mathbf{U}\_k = \\mathbf{U} \\mathbf{w}\_{u\_k}
> \\]
> where \\( \\mathbf{w}\_{a\_k} \\in \\mathbb{R}^n \\) and \\( \\mathbf{w}\_{u\_k} \\in \\mathbb{R}^m \\) are the \\( k \\)-th transformation vectors from the matrices \\( \\mathbf{W}\_A \\) and \\( \\mathbf{W}\_U \\), respectively. \\( \\mathbf{A}\_k \\in \\mathbb{R}^q \\) represents the transformation of the original attribute matrix \\( \\mathbf{A} \\) (size \\( q \\times n \\)) onto the \\( k \\)-th canonical direction, using the transformation vector \\( \\mathbf{w}\_{a\_k} \\in \\mathbb{R}^n \\). It results in a vector of size \\( q \\times 1 \\) that contains the transformed values for each user for the \\( k \\)-th component. Similarly, \\( \\mathbf{U}_k \\in \\mathbb{R}^q \\) represents the transformation of the original user embedding matrix \\( \\mathbf{U} \\) (size \\( q \\times m \\)) onto the \\( k \\)-th canonical direction, using the transformation vector \\( \\mathbf{w}\_{u\_k} \\in \\mathbb{R}^m \\). It also results in a vector of size \\( q \\times 1 \\) that contains the transformed values for each user for the \\( k \\)-th component.
>
> The goal of CCA is to maximise the Pearson Correlation Coefficient (PCC) between these transformed representations, i.e., between \\( \\mathbf{A}\_k \\) and \\( \\mathbf{U}\_k \\), for each \\( k \\)-th canonical component. The correlation for the \\( k \\)-th canonical component, denoted as \\( \\rho\_k \\), is given by the formula:
> \\begin{equation}
> \\rho\_k = \\frac{\\sum\_{i=1}^q \\left( \\left( \\mathbf{A}\_k \\right)\_i - \\mu\_{\\mathbf{A}\_k} \\right) \\left( \\left( \\mathbf{U}\_k \\right)\_i - \\mu\_{\\mathbf{U}\_k} \\right)}{\\sqrt{\\sum\_{i=1}^q \\left( \\left( \\mathbf{A}\_k \\right)_i - \\mu\_{\\mathbf{A}\_k} \\right)^2 } \\sqrt{\\sum\_{i=1}^q \\left( \\left( \\mathbf{U}\_k \\right)\_i - \\mu\_{\\mathbf{U}\_k} \\right)^2 }}
> \\end{equation}
> where \\( \\mu\_{\\mathbf{A}\_k} \\) and \\( \\mu\_{\\mathbf{U}\_k} \\) are the means of the transformed attribute and user embeddings, respectively, and \\( q \\) is the number of users.
>
> To be continued

---

> > ### Author Response · Authors · 2025-03-14
> > **Response to reviewer GqpX, continued**
> >
> > In matrix form, we can express this objective as maximising the correlation between the transformed matrices \\( \\mathbf{A} \\mathbf{W}\_A \\) and \\( \\mathbf{U} \\mathbf{W}\_U \\). This is formalised as:
> > \\begin{equation}
> > \\rho = \\max\_{\\mathbf{W}\_A, \\mathbf{W}\_U} \\operatorname{corr} \\left( \\mathbf{A} \\mathbf{W}\_A, \\mathbf{U} \\mathbf{W}\_U \\right)
> > \\end{equation}
> > where the correlation is maximised across the projection matrices \\( \\mathbf{W}\_A \\) and \\( \\mathbf{W}\_U \\), and the result is a set of \\( K \\)-canonical correlations \\( \rho\_1, \rho\_2, \\dots, \\rho\_k \\) that describe the relationship between the attribute embeddings and the user embeddings for the entire dataset.
> >
> > Thus, by computing the canonical correlations \\( \\rho\_K \\) for each component, we obtain insights into how well the attribute embeddings and computed user embeddings are aligned in terms of their underlying structure.
> >
> > > Furthermore, the Authors could consider rephrasing their contribution more in terms of the quantitative explorative findings they have, and less on the effective statistical significance of embeddings, which is rather reasonable and maybe less surprising / interesting.
> >
> > Thank you for your suggestion, we have revised our contribution in a more quantitative way (Our Contributions in page 2). We will also avoid the comment on the semantic space for the random baseline in SBERT.
> > > In Figure 5, why do the authors consider the second canonical component of CCA? This could be suspicious, why was this preferred on the 1st component?
> >
> > The first canonical component is more strongly correlated with age, and the second is more strongly correlated with gender. We choose to showcase the second component because gender is divided into two categories whereas age is divided into more than two categories, making the correlation less immediate.
> > Although the first component does capture male gender information, its representation is less distinct. For completeness, we have included the figure for the first component in the Appendix J.
> > > The Author could consider projecting the embeddings in a lower dimensional space to preserve the statistical properties of the embeddings while benefitting numerics (https://en.wikipedia.org/wiki/Johnson%E2%80%93Lindenstrauss_lemma)
> >
> > We appreciate the suggestion of using dimensionality reduction techniques, such as those inspired by the Johnson–Lindenstrauss lemma. We will explore this as part of future work (page 13).

---

> > > ### Comment · Reviewer_GqpX · 2025-03-17
> > >
> > > I thank the Authors for their rebuttal. In particular, I appreciate the additional Section A, which clarifies the definition of $\rho$, and the revision of the introduction / related work section.
> > >
> > > I read the other 2 reviews. While I do find this work incremental with respect to the work "On compositionality in data embedding" (Zhaozhen et al. 2023), as pointed out by Reviewer yF7A, I agree with the Authors that this submission technically adheres to TMLR policies.
> > >
> > > A mentioned weakness is the random shuffling as baseline to investigate embedding compositionality. In my opinion, while weak, this is a statistically significant baseline, and this is what this work aims to deliver to the reader. It is true that in specific contexts a more "effective" (in some sense) baseline can be considered, but I do not believe this to be the point here...
> > >
> > > Finally, I see a disagreement with Reviewer H4fb on the effective meaning of "Compositionality". I am fairly not familiar with the nomenclature in this sub-field, and thus I do not share the same concerns as Reviewer H4fb. At the current stage, I find the the arguments of the Authors convincing enough.

---

### Review · Reviewer_H4fb · 2025-02-24

**Summary Of Contributions:**

This paper applies the methodology of **Latent Semantic Analysis** to contextual embeddings. More precisely, (1) for sentence embeddings calculated from BERT-like models, it found that there exist linear sub-spaces in the latent embedding space that are well-correlated with orthogonal concepts, such as [_location_, _theater name_, _genre_] for movie-related dialogues, [_subject_, _verb_, _object_] for syntax of sentences;
(2) for knowledge graph embeddings calculated from (_user_, _rating_, _movie_) data, it found linear sub-spaces corresponding to [_gender_, _age_] of user demographics; and (3) for word embeddings calculated by word2vec, it found sub-spaces correlated to several different morphemes.

The methods are based on linear dimension reduction, such as **Canonical Correlation Analysis**; the authors call their methods "_Correlation-based Compositionality Detection_" and "_Additive Compositionality Detection_", which I believe are overreaching and inaccurate. So is their conclusion that "_embeddings including those from large and non-linear models, retain a surprising degrees of additive compositional structure_" -- what the authors actually did are quite far from what I expected by reading this claim. Please see the discussions below.

**Audience:**

Yes

**Broader Impact Concerns:**

None.

**Claims And Evidence:**

No

**Requested Changes:**

I'm not willing to accept that this paper is about Understanding Compositionality in Data Embeddings -- it's not about compositionality.

If the authors believe that this work is more closely related to compositionality than I understood, discussions are welcome -- and let's also discuss some more theoretical related works that are currently not cited in the paper, for example:

* Mitchell, Lapata (2010) Composition in distributional models of semantics. Cognitive Science

* Coecke, Sadrzadeh, Clark (2010) Mathematical foundations for a compositional distributional model of meaning. Linguistic Analysis

* Tian, Okazaki, Inui (2017) The Mechanism of Additive Composition. Machine Learning

* Arora, Li, Liang, Ma, Risteski (2018) Linear Algebraic Structure of Word Senses, with Applications to Polysemy. Transactions of the Association for Computational Linguistics

* Boleda (2020) Distributional Semantics and Linguistic Theory. Annual Review of Linguistics

**Strengths And Weaknesses:**

### Strengths:

The authors have done many experiments across different datasets and models. Despite the overreaching name, the experiments are meaningful.

### Weaknesses:

I believe the authors have a wrong understanding regarding (additive) compositionality. I believe "linearity" is probably a better name: "Understanding Linearity in Data Embeddings"; "Correlation-based Linear Sub-space Detection" and "Linearity Detection" would probably better describe the intuition.

Overall, the paper is about finding latent structure in the embedding space that can be represented by linear sub-spaces, even if the embeddings are calculated by non-linear, large neural networks. This is understandable because both the model design and training of deep neural networks are heavily based upon linear algebra -- the non-linearity is added in a somewhat controlled manner (e.g. the ReLU activation function and the Residual Connection) and usually the hidden vectors are robust to scaling changes (e.g. LayerNorm). But this is not compositionality; it is called the **Distributional Hypothesis**, that "conceptes" naturally arise as clusters discovered from distribution patterns of data, and "embeddings" trained on the data are such a multi-aspect clustering method. This hypothesis, is only about **one embedding vector**, and its relation to semantic concepts; so is this paper.

The study of compositonality goes one step further. It is about how the embeddings of constituent words are "composed" to make the embeddings of the context. So there are **multiple embedding vectors** in consideration. Specific to the settings of BERT-based sentence embedding, the contextualized embeddings read out from the final layer of BERT is the already composed vector that represents context; and how it is composed from the constituent word embeddings coming from the input first layer, seems pretty clear -- it is calculated by the Transformer architecture, and the model weights in the Transformer are learned from pre-training MLM loss. -- However, can we find general relations among the contextualized embeddings of the final layer and the constituent word embeddings from the input layer, that do not depend on the random initialization of the Transformer model and any specific model weights, but instead are characterized by the syntactic / semantic relations between the context and the constituent words? -- This is the difficult and important problem of compositionality, which is also related to the interpretability of neural networks.

One might think that an embedding in consideration (e.g. the sentence vector of a movie-related dialogue), having projections onto linear sub-spaces that correlate to constituent concepts (e.g. location, theater name, genre), as derived from the Distributional Hypothesis, is enough to imply some kind of "additive compositionality" because the linear algebra is about addition. This is a common mistake. In order to have additive relations between the embedding in consideration (=sentence vector) and the **embedding vectors** of the concepts (which, in this case, could be the cluster center of all contextualized embeddings in the dataset that represent the concept), one has to find out what happens **outside of the linear sub-space** that correlates to the concept -- because usually there is no guarantee that the embedding vector of the concept lives inside the linear sub-space! One example of such an analysis can be found in _Tian et al. (2017) The Mechanism of Additive Composition_, which shows that the component of "outside the linear sub-space" can somehow cancel out, in specific cases.

Now let's return to some specific settings of the experiments in this paper. In the sentence embeddings experiments, the sentence vector is obtained by mean-pooling the final layer contextualized token embeddings; so it's tautological to say that the mean-pooling is an addition of the final layer embeddings, and the experiments just show that the final layer contextualized embeddings align well to abstract concepts such as location and genre. This, my point again, is a verification of the Distributional Hypothesis; but it's not even related to the interpretability of the Transformer model -- you have some contextual embeddings which align well with semantic concepts; the contextual embeddings are produced by a black-box called Transformer, but we don't get any more interpretable insights into the Transformer black-box.

The last one is a minor criticism: In Appendix E.0.1, it says 'In this experiment, we replace some multi-token words with single token words. Specifically, we change “hamster” to “bear”, “hedgehog” to “fox”, “bookshelf” to “book”.' -- why doing this? There does not seem to be any technical difficulties regarding multi-token words, and the word replacement is weird. The cost to re-run this experiment in a more reasonable setting should be low, given that there is no human annotation involved and the computation required is just to run BERT inference, which can be done in a laptop.

---

> ### Author Response · Authors · 2025-03-14
> **Response to Reviewer H4fb**
>
> Thank you for your thoughtful review and for engaging  with our work. We appreciate the oppurtunity to discuss these ideas further and look forward to a productive exchange.
> > I believe the authors have a wrong understanding regarding (additive) compositionality. ‘linearity’ is probably a better name: ‘Understanding Linearity in Data Embeddings’...
>
> Thank you for the suggestion. We're open to changing the title to "Linearity," but we'd appreciate a brief discussion to ensure it fully reflects our approach. We would start by looking at the linguistic theory that underlies the principle of compositionality: the idea that the meaning of a complex expression is determined by the meanings of its parts and the rules used to combine them. This notion is central to Montague semantics (Stanford Encyclopedia of Philosophy [1]). In this paper, our goal is to understand compositionality within the embedding space, where the complex meanings of words, sentences, or graph nodes emerge from the combination of simpler components.
>
> **Compositionality VS Linearity** Our additive compositionality approach is fundamentally rooted in the linear system theory, as the **superposition** principle (see [2]) enables us to decompose embeddings into distinct component vectors that can then be recombined to reveal underlying relationships. In linear systems, superposition means that the overall response of a system to a combination of inputs is equal to the sum of the responses to each individual input. In this way, linearity and additive compositionality are effectively the same: both describe how the properties of a whole can be derived from the sum of its parts.
>
> Word2Vec [3], for instance, set the goal for understanding compositionality by showing that vector arithmetic can capture meaningful relationships (king - man + woman ≈ queen). This pioneering work has inspired further research on decomposing embeddings to discover compositional structures.
>
> We extend this line of inquiry by investigating compositionality across various data embeddings and evaluation metrics. Our “additive compositionality detection” method decomposes embeddings into constituent component vectors and then recombines them, clearly evidencing an additive signal that supports compositionality.
>
> Indeed, prior work, such as Mitchell and Lapata (2010) and Coecke, Sadrzadeh, and Clark (2010), constructs representations by combining parts into a whole. In contrast, we solve the same task inversely by decomposing holistic representations into their parts. **Thus, the task is the same, and we build on these approaches to offer a united method for understanding compositionality from both directions (We will give an example later).** Besides this, we would also argue that there are precedents for learning compositional representations via decomposition.
>
> **Moreover, previous work has demonstrated that compositional representations can be obtained by decomposing holistic vectors into parts.** Examples are Maillard and Clark (2015), Wijnholds et al (2020), Nazir and Sadrzadeh (2024). And indeed this is done in Baroni and Zamparelli (2010) (BZ) as well. Admittedly, in these papers, usually one component is taken to be fixed and then the other component is learned.  For example in BZ holistic adjective-noun vectors are learnt, noun vectors are learnt, and then adjectives are learnt as the matrix that solves the linear regression problem. However in Lewis et al (2024), nouns, relations, and adjectives are simultaneously learnt as decompositions of holistic vectors which in this case are image representations. In all of these papers the rules of composition are more complex than in the current work, however we would argue that our approach of decomposing into component vectors and then recombining to construct unseen sentences/words/graph nodes is analogous to the approaches taken in these previous works.
>
> To be continued
>
> [1] Montague Semantics. Stanford Encyclopedia of Philosophy.
>
> [2] Superposition Principle. Wikipedia.
>
> [3] Mikolov, T., Chen, K., Corrado, G., & Dean, J. (2013). Distributed Representations of Words and Phrases and their Compositionality. Advances in Neural Information Processing Systems 26 (2013).
>
> [4] Baroni, M., & Zamparelli, R. (2010, October). Nouns are vectors, adjectives are matrices: Representing adjective–noun constructions in semantic space. EMNLP 2010.
>
> [5] Martha Lewis, Nihal Nayak, Peilin Yu, Jack Merullo, Qinan Yu, Stephen Bach, and Ellie Pavlick. 2024. Does CLIP Bind Concepts? Probing Compositionality in Large Image Models. Findings EACL 2024.
>
> [6] Maillard, J., & Clark, S. (2015). Learning Adjective Meanings with a Tensor-Based Skip-Gram Model. CoNLL 2015.
>
> [7] Gijs Wijnholds, Mehrnoosh Sadrzadeh, and Stephen Clark. 2020. Representation Learning for Type-Driven Composition. CONLL 2020.
>
> [8] Saba Nazir and Mehrnoosh Sadrzadeh. 2024. How Does an Adjective Sound Like? Exploring Audio Phrase Composition with Textual Embeddings. CLASP 2024.

---

> > ### Author Response · Authors · 2025-03-14
> > **Response to Reviewer H4fb, continued**
> >
> > > I believe the authors have a wrong understanding regarding (additive) compositionality. I believe ‘linearity’ is probably a better name: ‘Understanding Linearity in Data Embeddings’; ‘Correlation-based Linear Sub-space Detection’ and ‘Linearity Detection’ would probably better describe the intuition.
> >
> > We agree that linearity is a fundamental aspect of our analysis and indeed plays a central role. However, our work distinguishes between merely identifying linear correlations (which is necessary, if there is no correlation, then we cannot decompose it at all) and demonstrating that additive decomposition (i.e., recombining the extracted components) faithfully reconstructs the holistic embedding. This recombination is the key that confirms additive compositionality, going beyond simply observing linear structure.
> > > Overall, the paper is about finding latent structure in the embedding space that can be represented by linear sub-spaces, even if the embeddings are calculated by non-linear, large neural networks. … But this is not compositionality; it is called the Distributional Hypothesis…
> >
> > While our methods do reveal latent linear structure, we argue that this structure is precisely what enables compositionality. The emergence of linear sub-spaces reflects the distributional properties inherent in the data. Our approach goes further by showing that these linear components can be additively recombined to recover the original, holistic representations (the leave-one-out experiment)—thereby operationalizing the concept of compositionality within the embedding space.
> >
> > While the Word2Vec method involves combining independently learned representations, our approach reveals an intrinsic compositional structure already present in the embedding space. The linear subspaces we uncover act as meaningful components that can be recombined to reconstruct full representations (as shown in our leave-one-out experiment). Thus, rather than imposing compositionality externally, our method makes explicit the compositional properties inherent in the data’s structure. This suggests that deep models inherently encode compositional relationships as part of their learning process, and our method makes these relationships explicit by identifying and recombining meaningful subspaces.
> >
> > > One might think that an embedding in consideration … is enough to imply some kind of ‘additive compositionality’ … In order to have additive relations between the embedding in consideration and the embedding vectors of the concepts … one has to find out what happens outside of the linear sub-space …
> >
> > Not sure if we understand this comment correctly. We agree that simply observing a linear correlation is not sufficient to establish additive compositionality. This is why our two-step approach: first detecting the correlation (using methods like CCA) and then performing an explicit additive decomposition: ensures that we validate the presence of an additive signal. The fact that we can decompose and then accurately reconstruct the original embedding supports the claim that the additive component is both meaningful and robust.
> > > Now let's return to some specific settings of the experiments in this paper. In the sentence embeddings experiments... and the experiments just show that the final layer contextualized embeddings align well to abstract concepts …
> >
> > First of all these concepts should not be expected to emerge simultaneously, since we do not use any of the explicit concepts for training. Secondly, the point that we are making is that that information is present in the original embeddings in an additive format. This is not obvious and this goes beyond the basic assumption that sentence embeddings merely contain some information about their constituent concepts. Take the case of the sentence embeddings. In the case of sentence embeddings, the embedding that we deconstruct consists of a weighted sum of (1) the residual from the MHA layer, which is a weighted sum of word embeddings, and (2) this residual passed through a nonlinear MLP. Moreover, the word embeddings thus combined will be slightly different in each sentence processed.
> >
> > > I'm not willing to accept that this paper is about Understanding Compositionality in Data Embeddings — it's not about compositionality. If the authors believe that this work is more closely related to compositionality than I understood, discussions are welcome …
> >
> > We respectfully maintain that our work does indeed contribute to understanding compositionality in data embeddings. **By demonstrating that embeddings can be decomposed and then recombined additively—mirroring the phenomena observed in Word2Vec and related works—our approach aligns with established definitions of compositionality**. We welcome further discussion and will incorporate the suggested theoretical references to provide a richer context for our contributions in the revised manuscript.

---

> ### Author Response · Authors · 2025-03-14
> **Response to Reviewer H4fb: Our work and Compositionality---an example**
>
> One of the most important reasons for researching compositionality is its role in enabling the creation and understanding of new combinations of components that generalize to novel cases. For example, in language, understanding the meaning of "eco-friendly transportation" does not require prior exposure to the phrase; rather, we can infer its meaning by combining the known concepts of "eco-friendly" and "transportation." Similarly, in embedding spaces, if we can represent and reconstruct new concepts by composing their underlying components, it demonstrates that the model has learned a structured and generalizable representation, reinforcing the fundamental idea of compositionality. Our **leave-one-out** experiments (Figures 2, 4, 7) align with this principle by demonstrating that embeddings can be decomposed and recombined in a structured way.
>
> For example, in our knowledge graph embedding, we test whether we can reconstruct an embedding based on known components. We will start from a toy example.
>
> Given the embeddings and identity labels of 50-male, 20-female, and 20-male, we investigate whether we can accurately reconstruct the embedding for 50-female. The answer is yes, and we validate this result statistically (Figure 4b).
>
> We also test the inverse case. Given the same set of embeddings and an additional embedding for 50-female, we ask whether we can correctly predict its identity—determining that it represents a 50-year-old female. Again, the answer is yes, and we prove this statistically (Figure 4c).
>
> A similar structure appears in word embeddings. Given the embeddings of physically, physicalness, and studiously, we examine whether we can predict the embedding of studiousness (and the inverse way). Our results confirm that this prediction is successful, demonstrating that compositionality holds in language representations as well (Figure 7 a,b,c).  Note that there are way more combinations in our experiments, and there are more metrics as well.
>
> **These findings confirm that we can systematically produce and understand new combinations of component embeddings**, such as 50-female or studiousness (and vice versa for the identity prediction), which aligns with the core goal of compositionality.
>
> Indeed, one might argue that the difference between 50-male and 50-female is not the same as the difference between 20-male and 20-female, suggesting that compositionality behaves differently across contexts. (this observation relates to the linear subspaces governing how attributes interact) However, the **leave-one-out** method provides a robust, quantitative way to measure compositionality by showing that embeddings can be decomposed and reconstructed in a structured, linear way.
>
> Thus, our methods and experiments clearly fall under the concept of compositionality, as they offer a systematic approach to reconstructing embeddings and identities from their components.

---

> ### Comment · Reviewer_H4fb · 2025-03-25
> **Thanks for the response**
>
> I have carefully read the authors' engaging response. Thanks for the discussion. I want to make 4 points in my comment.
>
> **0.** Let's distinguish the following:
>   * A "principle component", as in the name Principle Component Analysis (PCA), of a vector space, in the mathematical sense;
>   * A "concept", a mental representation of a thing or class of things that serves as the foundation of our understanding and knowledge;
>   * A "linguistic constituent", which is a word or a group of words that carry a certain syntactic role in a sentence;
>   * A "semantic feature" or "semantic component", which is considered as the basic building blocks of meaning;
>
> The word "composition" and "compositionality" have very specific meaning in linguistics. It means the **syntactic** process of combining linguistic constituents into larger units of text, like sentences. One believes that "meaning" is constructed along the process as well, through a "syntactic-semantic" interface. (Example: _Mary saw a dog._ This sentence describes an event, while each constituent word only carries partial information of that event: the word "dog" says the object, the word "saw" describes the action and specifies the tense (i.e. it already happened), and the word "Mary" says the subject. Composition is the process that the meaning of the whole sentence is constructed through the combination of these words.)
>
> This should not be conflated with the idea of "semantic decomposition", that every concept can be broken down to a set of basic building blocks called "semantic features". For example, the word "run" means "move", "fast", and "on foot". It is the idea that concepts could be organized as a hierarchy, and more specific concepts could be described by taking the intersection of multiple, more general concepts.
>
> So "linguistic composition" and "semantic decomposition", these are two different ideas.
>
> In the era of deep learning, vector representations of meaning are mainstream. One usually finds that low-dimensional embeddings learned from probabilistic distributions of word occurrences can align well with some concepts, which justifies another, different understanding of meaning: The belief of "Distributional Hypothesis", that meaning arises as the principle components of some distribution correlation matrix, among word and usage.
>
> I'm laying down these three different ideas about "meaning", not for policing different camps, but for setting up the stage. After all, it's the potential of combining these three ideas that makes the research topic interesting. More specifically, we already know that embeddings, or low-dimensional vectors, can align well with concepts; could we go further and align _algebraic operations_ of vectors, to the _construction process_ of meaning, as described by either linguistic composition or semantic decomposition?
>
> My opinion is that the experiments done in this paper do not go beyond (re)verification of the Distributional Hypothesis, that principle components of correlation matrices can align well with meaningful concepts.
>
> Previously, the authors claim that they were detecting "additive compositional structure" in BERT embeddings, which as I understand, should be expected to show that the last layer embedding, as calculated from the first layer through a complicated neural network of BERT, can be approximated by simple additions of the first layer embeddings, maybe according to the syntactic roles of the input words -- but obviously that's not what's done in this paper.
>
> In the responses, the authors have changed the focus to "semantic decomposition", claiming that their experiments verify the idea that every concept can be broken down to a set of basic building blocks or semantic features, and this process can be simulated by simple addition of vector representations. Now I'm going to explain why the experiments are not showing that, either.

---

> > ### Comment · Reviewer_H4fb · 2025-03-25
> > **(continued.)**
> >
> > **1.** Given an embedding, any list of concepts, and correlation data between the embedding and the concepts, the methods in this paper (i.e. CCA and alike) will find components that are "significant" to the correlation. But there is no guarantee that the found components will be aligned to **more fundamental, or basic building blocks** of semantic features.
> >
> > The list of concepts to be used is an arbitrary choice. It could be a randomly selected list of words in the vocabulary, a list of sentences sampled from corpora or generated by GPT-4, or images linked to captions -- as long as there is meaningful correlation, the methods in this paper will produce some principle components of the embedding space. Do you call all of them "compositionality detection"? If this is the 2010s, are you going to call it the "topic model" because the clusters found by the method somehow represent meaningful topics? -- The methods used in the paper do not, and cannot imply that the found principle components represent **more basic** concepts, or that there is a hierarchy of concepts at all.
> >
> > **2.** In the settings of this paper, vector representations of the list of concepts are not given at the beginning. It's after the singular value decomposition, and you say "let's assume that this concept is represented by this principle component, and voila: If we combine all the principle components, we can recover the original embedding!" -- This is like shooting the arrow first and drawing the target after.
> >
> > One usually needs to decide first which vector representations to use for the list of concepts; then, the experiments might be able to show something nontrivial. Or, if the vector representations of the list of concepts are obtained as a result of the method, one needs an additional step of manual evaluation to show that the vector representations can indeed align well with the concepts as understood by humankind. Currently I don't find such evaluation in the paper.
> >
> > **3.** Even if we accept the principle components as vector representations for the list of concepts, we don't have the vector yet. We only have a **direction** -- one still needs to decide the scale, or norm of the vector. For simplicity, let's assume that the scale of vector representations are all normalized to 1. Now if we take the sum (of equal weights) of the vectors of correlated concepts, there is no guarantee that we can recover the original embedding! -- If you allow arbitrary weights, that of course is possible, but what's the point? Every d linearly independent vectors can span a vector space of dimension d.
> >
> > What's outstanding about Mikolov's analogy task, is not only the simple operations of vector additions, but also the coefficients all set to 1. It's not so outstanding if one says _queen = 0.8 * king - 0.5 * man + 1.2 * woman_. Currently I don't see any careful evaluation or discussion about this issue in the paper at all. By the way, the name given to the analogy task is not "compositionality", but "linguistic regularity", as in _Linguistic Regularities in Continuous Space Word Representations_, NAACL 2013; that paper never used the word compositionality. In the NIPS paper _Distributed representations of words and phrases and their compositionality_, when they discuss compositionality, it was exclusively about the specific process of composing the meaning of a phrase by its constituent words.
> >
> > If the authors of this paper would like to continue the exploration of finding evidence of semantic decomposition in vector representations, the following
> >
> > * Arora, Li, Liang, Ma, Risteski (2018) Linear Algebraic Structure of Word Senses, with Applications to Polysemy. Transactions of the Association for Computational Linguistics
> >
> > might be a good starting point. They use the word "linear algebraic structure" instead of "additive compositional structure", and they actually discussed the addition of concrete word vectors, with linguistic evaluations toward understanding polysemy. It could be a good framework to follow for research in this direction. Moreover, I believe it is important for the authors to clearly state their claims, without vaguely referring to words like "composition" or "decomposition".

---

> ### Author Response · Authors · 2025-03-25
> **Response to Reviewer H4fb**
>
> We thank you for your detailed and thoughtful comments. We appreciate your careful review of our work and the opportunity to clarify our methods. We understand that there may be some misunderstandings regarding our approach. In our revised manuscript, we will rename our first method as "linearity detection" to more accurately reflect its purpose of verifying the presence of linear correlations in the embedding space. For our second method, we will provide additional explanations and clarifications. Specifically, we will clearly distinguish the two methodological components as follows:
>
> **1. Correlation-Based (Linearity Detection) Method:**
>
> - In this approach, we use Canonical Correlation Analysis (CCA) to identify significant linear correlations between graph/sentence/word embeddings and a candidate set of demographic attributes/concepts and etc.
>
> - We agree that this method, by itself, is not **directly** a test of compositionality. In response to your comments, we will revise the terminology to “linearity detection” rather than “compositionality detection.”
>
> - The **purpose** of this step is to verify that a linear signal exists in the embedding space. As you rightly note, if we cannot even detect a robust linear correlation (e.g., via the principal components from CCA), there would be no basis for any further strict decomposition.
>
> **2. Linear Decomposition-Based Additive Compositionality Detection:**
>
> - This second method is **entirely distinct from** the first and does **not** rely on principal component analysis.
>
> - In our approach, we posit that the final embedding (produced by BERT/GPT/LLAMA/word2vec/KG) can be approximately decomposed as an additive combination of “concept/demographic embeddings.”
>
> - We formalize this via a linear system: $$AX=U$$
>  where $U$ is the sentence/node (user)/word embedding, $A$ is a multi-hot (binary) identity coefficient matrix (with entries set to 1 when a particular concept is present), and $X$ is the matrix of concept/demographic embeddings to be solved for. – This formulation is analogous to Mikolov’s analogy task, where the vector additions are performed with fixed coefficients (all set to 1). In our case, the multi-hot matrix $A$ ensures that every token or identity present in a sentence contributes equally with a coefficient of - For the sake of simplification, when we mention "sentence" below, it refers to a sentence, word, or node in a graph.
>
> **Leave-One-Out Experiment:**
>
> - We remove the identity information and embedding (i.e., the corresponding row in $A$ and $U$) for **one sentence** and solve for the concept embeddings $X$ using the **remaining data**.
>
> - We then reconstruct the unseen sentence embedding using the computed $X$ and the original identity indicators from $A$ (also in an inverse way, reconstruct the identity from the computed $X$ and sentence embedding $U$).
>
> - Successful reconstruction demonstrates that the sentence embedding is indeed an additive combination of the concept embeddings (we show this by comparing with the random shuffled baseline).
>
> **Random Shuffling Control:**
>
> - To further validate our approach, we randomly shuffle the sentence–identity pairings and attempt the same reconstruction.
>
> - The failure to accurately reconstruct in the shuffled setting confirms that the additive structure is not an artifact but is intrinsic to the correct pairing between the sentence embeddings and the concept identities.
>
> - Figure 2,4,7 all beat the random baseline.
>
> **Point-by-Point Response**
>
> **1. On the Role of Principal Components in the CCA-Based Method:**
> > Simply finding out the principal components in CCA is not enough…”
>
> - We agree that merely extracting principal components does not establish compositionality.
>
> - Accordingly, we will rename this method to “linearity detection” to reflect that its role is solely to verify the presence of linear correlations in the embedding space—a necessary precursor for our subsequent decomposition.
>
>
> **2. On the Sequential Order of Concept Identification and Reconstruction:**
>  > “…this is like shooting the arrow first and drawing the target after.”
>
> - Our method explicitly sets up the linear system $AX=U$, where the concept embeddings $X$ are derived from the intrinsic structure of the final sentence embeddings.
>
> - The **leave-one-out** experiment is designed to test whether the additive combination of these derived components can faithfully reconstruct an **unseen** sentence embedding (We are predicting an unseen case).
>
> - The additional shuffling experiment further confirms that correct pairing is essential for accurate reconstruction, thereby supporting the validity of the additive structure (please refer to the comments of reviewer GqpX’s statistically significant baseline).
>
> To be continued

---

> ### Author Response · Authors · 2025-03-25
> **Point-by-Point Response, continued**
>
> **Point-by-Point Response, continued**
>
> **3. On the Setting of Coefficients (Uniform Weights):**
>  > “What's outstanding about Mikolov's analogy task, is not only the simple operations of vector additions, but also the coefficients all set to 1.”
>
> - In our formulation, the coefficient matrix $A$ is a multi-hot indicator matrix where every concept present in the sentence contributes with a coefficient of 1 (details in Appendix B).
>
> - This is directly analogous to Mikolov’s analogy task.
>
> - The fact that we achieve accurate reconstruction using these fixed, uniform coefficients underscores the presence of a meaningful additive compositional structure in the sentence embeddings.
>
> **4. On the Nature of Additive Compositionality in Our Work:**
> > “…the experiments do not go beyond (re)verification of the Distributional Hypothesis.”
>
> - While our CCA-based method (now termed “linearity detection”) verifies the presence of linear correlations, our second method explicitly demonstrates additive compositionality.
>
> - By solving $AX=U$ with fixed coefficients and validating the reconstruction via leave-one-out and control experiments, we show that the final sentence embeddings are not arbitrary aggregations.
>
> - Rather, they are composed additively from underlying concept embeddings in a manner that is robust and intrinsic to the embedding space.
>
> **Clarifying Terminology:**
>
> The term “additive compositionality” here reflects the idea that these components, once identified, can be additively recombined to faithfully reconstruct the overall embedding. (There is no need for arbitrary or fine-tuned weights beyond the binary coefficients provided by $A$.) We do not claim that the final sentence embedding is the same as a simple sum of the pre-BERT token embeddings. Rather, our work shows that: even after the complex, non-linear transformations within BERT/GPT/LLAMA, the resulting sentence embedding can be decomposed into a linear combination of concept vectors.
>
> In summary, our paper employs a two-step approach:
>
> The first method (linearity detection via CCA) establishes that there is a robust linear correlation between the embeddings and candidate semantic features.
>
> The second method (linear decomposition-based additive compositionality detection) rigorously demonstrates that the final sentence embeddings can be reconstructed as an additive combination of concept embeddings, with uniform coefficients analogous to those in word2vec’s analogy tasks.
>
> We hope this layered approach could address your concerns. We appreciate your suggestions and will incorporate these clarifications and terminological revisions in the revised manuscript.

---

### Review · Reviewer_yF7A · 2025-03-05

**Summary Of Contributions:**

This paper studies whether embeddings at different granularities (e.g., word, sentence, and graph-level representations) can be decomposed as additions of representations of interpretable concepts/attributes. The authors explore this question by using two previously proposed methods for measuring a representation’s compositionality, namely CCA and “additive compositionality detection”, to assess the embeddings learnt by several transformer-based and knowledge-graph-based models at different representational levels and training stages. Their findings show that (1) sentence embeddings exhibit additive compositionality, allowing decomposition into conceptual components, (2) word embeddings capture both semantic and morphological structures, supporting decomposability into roots and suffixes,  (3) knowledge graph embeddings can additively encode demographic attributes, revealing key data structures without explicit supervision, and (4) compositionality is most prominent in intermediate layers and increases with early training before stabilizing.

**Audience:**

Yes

**Broader Impact Concerns:**

I do not believe there are any broader ethical implications from this work that would merit a Broader Impact Statement.

**Claims And Evidence:**

Yes

**Requested Changes:**

Below, I list some potential changes for each section and their importance in securing my recommendation for acceptance. If time is limited, please focus on all critical requests, followed by all major requests. If I misunderstood something at any point, just let me know, as this is always possible (apologies in advance if that’s the case!).

### Section 1 (Introduction)

- **(Minor, Nit)** The relationship `king - man + woman = queen` is more accurately characterized as `king - man + woman` **≈** `queen`
- **(Minor, Potential Typo)** For consistency, “Sentence embedding, performed…” should probably be “Sentence embeddings, performed…”
- **(Minor, Typo)** “In examining sentence embeddings” should probably be “When examining sentence embeddings”

### Section 3 (Evaluating Compositionality of Data Embeddings)

- **(Critical, Proper Attribution)** The SGD dataset should be correctly attributed to the authors with a correct citation (a link to the dataset is not a proper citation/attribution). Please look into that repository and cite the work as requested by the authors in that repository’s README.
- **(Minor, Typo)** “Behaviour of SBERT …” should be “The behaviour of SBERT …”
- **(Critical, Statistical Analysis)** Given that you are reporting mean values in Table 1, can you please include the appropriate error bars for those statistics? This would allow readers to get an insight into the variance across the different concepts/attributes. **This same comment applies to other quantitative tables/results in this paper.**
- **(Minor, Typo)** “Test statistics for SBERT embedding decomposition.” should probably be “Test statistics for SBERT’s embedding decomposition.”
- **(Critical, Question)** What does Figure 2 give us that Table 1 + errors would not already give us? I am trying to understand what I need to conclude from that figure; hence, it would be helpful to have some guidance on what I should look for in that figure that I cannot find from previous parts of the paper. If it is otherwise redundant, then I would reconsider its inclusion in the main body of the paper. A similar point applies to Figure 4.
- **(Minor, Question)** Is the hypothesis that “semantic information is better encoded at earlier layers” the only plausible explanation for the observed difference in compositionality metrics for the last layer of SBERT? If so, we should observe similar results across different sentence transformer models with different sizes and depths. Is that the case? If this is not the case, then there may be another reason behind this result which could also be attributed to an issue in the original design of the metric being considered.
- **(Critical, Question)** What is the intuition behind the (very) high real and random cosine similarities observed in Table 3 even before any training has been made? These very high values make me uneasy about trusting high values of this metric for really capturing a useful signal in the absence of a baseline to compare them against.
- **(Minor, presentation)** I would recommend making the text in some of the figures less cluttered and larger as otherwise it requires significant zooming to read (e.g., Figures 1, 3, and 5).

### Appendix E (Decomposing Sentence Embedding into Subject, Verb and Object)

- **(Minor, Typo)** “thier” should be “their”
- **(Critical, TMLR Policy Potential Conflict)** The fact that a lot of work from [3] is directly used here (although properly cited) may be in conflict with TMLR’s policy of not allowing papers that “reuse the authors’ own prior work”. I am just flagging this for the AE to see but I apologize in advance if this is a misunderstanding of the policy and this work vis-a-vis [3]. If this is a misunderstanding, please ignore this concern.

### Appendix F (Word Embeddings)

- **(Minor, Typo)** There is a missing equation reference at the end of page 24.

### General Questions

- **(Critical, Question)** Could the authors please clarify how this work adheres to the TMLR policy requiring all contributions to be new (e.g., not extensions of previously published works) with respect to [3]? In particular, a lot of the experiments, and the methodology, of this submission seem to be an extension of [3], with some of the plots and text of this work and of [3] sharing a lot of commonalities. My understanding is that this work really expands the empirical observations of that paper (by looking at different models, training elements, and representations within a model), yet these all seem to be mostly immediate extensions of the previous work that, without an accompanying deep/non-trivial analysis, may be on the boundary of what is considered substantial advances w.r.t. previous work. Unfortunately, as the paper is structured at the moment, I am not entirely convinced that the analysis of these experiments provides substantial novel contributions that were not expected from the work we already saw in [3].
- **(Critical, Question)** Could the authors please comment on my concern regarding the actionability or key takeaways one would take from this work vis-a-vis existing works such as [3]? Besides confirming that the results of [3] can be seen in other domains/models, what can one do with these results/methods/tools/evaluation techniques that could actually help one make sense or work better with a group of embeddings? Making this extremely clear is key given that interpretability is used as an explicit motivation for this work while most of the results in this paper seem to suggest that previous results generalize across different areas, but the lack of actionability of those previous works/results remains.
- **(Critical, Question)** Regarding the selection of the random permutation baseline in this evaluation, are there any other existing decomposition approaches you could use as baselines for this? Perhaps within the family of topic or dictionary learning? I am not entirely convinced that the random baseline is a strong enough baseline for one to use to claim decompositionality in a learnt representation. Perhaps there is a simple toy experiment that one can design to show that when representations are NOT decompositional, then the metrics studied in this work and the random baseline are not (statistically) significantly different from each other? If so, then that would definitely add enough evidence on my side to convince me that maybe this is a good enough baseline for one to build evidence for decompositionality really occurring.

## References

- [1] Mikolov, Tomas, et al. "Distributed representations of words and phrases and their compositionality." Advances in neural information processing systems 26 (2013).
- [2] Guo, Zhijin, et al. "EXTRACT: Explainable Transparent Control of Bias in Embeddings." CEUR-WS (2023).
- [3] Xu, Zhaozhen, Zhijin Guo, and Nello Cristianini. "On compositionality in data embedding." International Symposium on Intelligent Data Analysis. Cham: Springer Nature Switzerland, 2023.

**Strengths And Weaknesses:**

Thank you so much for submitting this work! I enjoyed reading this paper, particularly its experimental section. Before I begin, I would like to point out that although I am comfortable and familiar with areas within representation learning, some of the areas touched in this paper (e.g., graph representation learning and compositionality in sentences) are areas that I am tangentially familiar with. As such, there is a non-zero probability that I might’ve missed key works or points when evaluating this paper.

That being said, below are what I believe are this paper’s main strengths, followed by what I believe are some of its weaknesses:

## Strengths

1. **[Significance] (Major)** The study of representational compositionality is a very active and important component in several areas of AI, from interpretability to debiasing, NLP, and neuro-symbolic modeling. As such, I believe the main empirical results in this paper may be of interest to a broad and large community.
2. **[Quality and clarity] (Major)** The paper is very well written and easy to follow. The experiments, metrics, and setups are really easy to understand and very well explained. I sincerely thank the authors for this.

## Weaknesses

In contrast, I believe the following are some of this work’s limitations:

1. **[Improvements over previous iteration] (Critical)**  I went over the previous iteration of this work, together with its review, and I believe the authors did a good job at incorporating the stylistic-and-evaluation-related feedback from that previous submission. However, I am still worried with the widely unaddressed concern expressed in that submission regarding this work being an extension of previous existing published work (e.g., [2] and **particularly** [3]). I am raising this point so that the AE can look into it, but I apologize if I misunderstood this. If the AE considers the level of contributions of this work to go beyond the bar set in TMLR to mark a work from being “an extension” to being “entirely novel”, then please ignore this comment.
2. **[Originality] (Minor, but related to the point above)** The key methodological ideas of this paper have been previously explored by others (e.g., [1, 2, 3]), as rightly pointed out in the paper itself. However, this work extends this analysis further to other data and embedding domains and proposes some interesting experimental methodologies for the analysis of compositionality. As such, I do not believe the main contributions are novel in the methodological side but they are interesting and novel in the experimental and analysis side of things. As novelty is not a key consideration per TMLR guidelines, I will only mark this as a minor weakness as long as the AE determines that this work goes beyond what one would consider an extension of previously published material.
3. **[Impact] (Critical)** This work shows that some previously observed results on compositionality generalize across different models, training regimes, and representations. Although interesting and helpful, these results still lack a clear actionability/use case that shows how one can use/exploit these empirical observations to address existing issues (e.g., lack of interpretability) or improve upon any other dimension (e.g., better training, more decomposable representation learning, etc.).
4. **[Quality] (Major)** As reviewers of previous iterations of this work have expressed, I am not entirely convinced that a random baseline for the hypothesis testing is strong enough for one to jump to big conclusions from the observed results. As such, I would appreciate it if the authors could consider incorporating non-random or ground-truth baselines, perhaps in more controlled scenarios as a toy experiment(s).

## References

- [1] Mikolov, Tomas, et al. "Distributed representations of words and phrases and their compositionality." Advances in neural information processing systems 26 (2013).
- [2] Guo, Zhijin, et al. "EXTRACT: Explainable Transparent Control of Bias in Embeddings." CEUR-WS (2023).
- [3] Xu, Zhaozhen, Zhijin Guo, and Nello Cristianini. "On compositionality in data embedding." International Symposium on Intelligent Data Analysis. Cham: Springer Nature Switzerland, 2023.

---

> ### Author Response · Authors · 2025-03-14
> **Response to reviewer yF7A**
>
> We sincerely appreciate your detailed review and insightful suggestions, and we are truly grateful for your kind comments.
>
> > (Minor, Nit) The relationship king - man + woman = queen is more accurately characterized as king - man + woman ≈ queen
> Done, we have also corrected the typos in this paper.
> > (Critical, Proper Attribution) The SGD dataset should be correctly attributed to the authors with a correct citation (a link to the dataset is not a proper citation/attribution). Please look into that repository and cite the work as requested by the authors in that repository’s README.
>
> Done (page 6)
> > (Critical, Statistical Analysis) Given that you are reporting mean values in Table 1, can you please include the appropriate error bars for those statistics? This would allow readers to get an insight into the variance across the different concepts/attributes. This same comment applies to other quantitative tables/results in this paper.
>
> See the following answer for the next question, we plot the distribution of all 100 permutations in Figure 2, 4, and 6. We will give a detailed answer below on why we do this.
> > (Critical, Question) What does Figure 2 give us that Table 1 + errors would not already give us? I am trying to understand what I need to conclude from that figure; hence, it would be helpful to have some guidance on what I should look for in that figure that I cannot find from previous parts of the paper. If it is otherwise redundant, then I would reconsider its inclusion in the main body of the paper. A similar point applies to Figure 4.
>
> Table 1 reports mean values computed across 100 permutations of the sentence-concept pairs, whereas Figure 2 visualizes the full distribution resulting from these 100 permutations. Displaying all 100 permutations is necessary because we are employing a non-parametric statistical approach—specifically, a Monte Carlo sampling procedure—to directly estimate the p-value. Here, the p-value represents the probability that a high correlation (or test statistic value) could occur purely by chance under the null hypothesis. If this probability (p-value) is found to be less than 1%, we can reject the null hypothesis, or at least support the alternative hypothesis. Therefore, Figure 2 is essential as it provides visual evidence of this distribution and aids readers in intuitively understanding the statistical significance of our results.
>
> > (Minor, Question) Is the hypothesis that “semantic information is better encoded at earlier layers” the only plausible explanation for the observed difference in compositionality metrics for the last layer of SBERT? If so, we should observe similar results across different sentence transformer models with different sizes and depths. Is that the case? If this is not the case, then there may be another reason behind this result which could also be attributed to an issue in the original design of the metric being considered.
>
> As suggested by reviewer GqpX, we will avoid commenting on the semantic space in this case. However, there are some previous studies that claim: that pre-training objectives impose a significant task bias onto the final layers for multiple common Transformer language models [1]. A similar discussion can also be found in paper [2].
> [1] Carlsson, F., Gogoulou, E., Ylipää, E., Cuba Gyllensten, A. and Sahlgren, M., 2021. Semantic re-tuning with contrastive tension. In International Conference on Learning Representations, 2021.
> [2] Ethayarajh, K., 2019, November. How Contextual are Contextualized Word Representations? Comparing the Geometry of BERT, ELMo, and GPT-2 Embeddings. In Proceedings of the 2019 Conference on Empirical Methods in Natural Language Processing and the 9th International Joint Conference on Natural Language Processing (EMNLP-IJCNLP) (pp. 55-65).
> > (Critical, Question) What is the intuition behind the (very) high real and random cosine similarities observed in Table 3 even before any training has been made? These very high values make me uneasy about trusting high values of this metric for really capturing a useful signal in the absence of a baseline to compare them against.
>
> At 0k steps, the model consists of randomly initialized weights and embeddings. One reason is the Random Initialization Effects: Even though weights are randomly initialized, they follow structured distributions, leading to embeddings with similar statistical properties. Additionally, Layer Normalisation’s smoothing effect can sometimes artificially inflate cosine similarity between embeddings, particularly before training or in random settings. Positional encoding could also be a possible reason, BERT models use learned positional encodings, which are added to embeddings before any training occurs, introducing an additional structured component.
>
> To be continued

---

> ### Author Response · Authors · 2025-03-14
> **Response to reviewer yF7A, continued (new contributions)**
>
> > (Critical, TMLR Policy Potential Conflict) The fact that a lot of work from [3] is directly used here (although properly cited) may be in conflict with TMLR’s policy of not allowing papers that “reuse the authors’ own prior work”. I am just flagging this for the AE to see but I apologize in advance if this is a misunderstanding of the policy and this work vis-a-vis [3]. If this is a misunderstanding, please ignore this concern.
> General Questions
> > (Critical, Question) Could the authors please clarify how this work adheres to the TMLR policy requiring all contributions to be new (e.g., not extensions of previously published works) with respect to [3]? In particular, a lot of the experiments, and the methodology, of this submission seem to be an extension of [3], with some of the plots and text of this work and of [3] sharing a lot of commonalities. My understanding is that this work really expands the empirical observations of that paper (by looking at different models, training elements, and representations within a model), yet these all seem to be mostly immediate extensions of the previous work that, without an accompanying deep/non-trivial analysis, may be on the boundary of what is considered substantial advances w.r.t. previous work. Unfortunately, as the paper is structured at the moment, I am not entirely convinced that the analysis of these experiments provides substantial novel contributions that were not expected from the work we already saw in [3].
>
> Thank you for carefully reviewing our submission and highlighting this concern. Clarifying and addressing overlap with previous work was indeed a primary goal in the current submission. While the methodology builds upon [3], this paper presents entirely new experimental results, substantially extending the empirical scope by exploring new properties and relationships within word, sentence, and graph embeddings, as well as different models and training conditions.
>
> According to the TMLR guidelines, contributions can include "accounts of applications of existing techniques that shed light on the strengths and weaknesses of the methods," or "reproducibility studies of previously published results or claims." We believe that this work fits within these criteria, as it provides deeper empirical analyses and novel insights into the strengths and limitations of embedding methods, thereby significantly enriching our understanding beyond what was established in [3].
>
> The main contribution of this submission is analysing new properties and relations in word, sentence and graph embedding.
>
> Specifically, in the case of word embedding, we find out the correlation with WordNet and morphology information, and afterwards decompose the embedding into these components. This is completely new, and hasn’t been done previously.
>
> In the case of sentence embedding, we introduce a new experiment and a new dataset to decompose the sentence embedding into constituent concepts. The decomposition itself is different from previous work of decomposing sentence embedding into subject, verb and object since we are focusing on other aspects rather than the previous sentence syntax.
>
> In the case of the graph embedding, we test different scoring functions, and different training stages of the knowledge graph embedding. These are new experiments.
>
> (We did extend the previous work of sentence SVO decomposition by different settings, and put it in the appendix. If you believe this is a problem, we are happy to remove them from the appendix.)
>
> To be continuted

---

> > ### Author Response · Authors · 2025-03-14
> > **Response to reviewer yF7A, continued**
> >
> > > (Critical, Question) Could the authors please comment on my concern regarding the actionability or key takeaways one would take from this work vis-a-vis existing works such as [3]? Besides confirming that the results of [3] can be seen in other domains/models, what can one do with these results/methods/tools/evaluation techniques that could actually help one make sense or work better with a group of embeddings? Making this extremely clear is key given that interpretability is used as an explicit motivation for this work while most of the results in this paper seem to suggest that previous results generalize across different areas, but the lack of actionability of those previous works/results remains.
> >
> > Thank you for raising this critical question.
> > In the embedding of words a series of interesting phenomena have been observed, whose extension to other forms of data is of great practical interest. They include “compositionality”, that is, the property that the embedding of two words that have certain semantic or syntactic relations are related in a predictable manner, typically in an additive form (what we did). The presence of biases, such as gender bias, has also been observed in the same representation. Both of those can be explained through the “distributional” assumption, that is the idea that the meaning of a word depends on the statistical distribution of words related to it. Thus, deeply understanding how compositionality emerges across different training stages and embedding domains can directly inform strategies for identifying and mitigating biases, improving transparency and interpretability in real-world applications.
> >
> >
> > Understanding linear relationships within embeddings is also crucial for mechanistic interpretability—breaking down complex neural network behaviors into understandable, linear components. Such decomposition allows practitioners to diagnose, manipulate, and optimize networks more effectively. Therefore, our analysis of embedding linearity provides practical guidance for interpreting and improving model behavior in broader contexts beyond confirming previous observations.
> > > (Critical, Question) Regarding the selection of the random permutation baseline in this evaluation, are there any other existing decomposition approaches you could use as baselines for this? Perhaps within the family of topic or dictionary learning? I am not entirely convinced that the random baseline is a strong enough baseline for one to use to claim decompositionality in a learnt representation. Perhaps there is a simple toy experiment that one can design to show that when representations are NOT decompositional, then the metrics studied in this work and the random baseline are not (statistically) significantly different from each other? If so, then that would definitely add enough evidence on my side to convince me that maybe this is a good enough baseline for one to build evidence for decompositionality really occurring.
> >
> > We agree that it is not enough to only compare with the random baseline. Nevertheless, beating the baseline means that we do detect a signal of that explaining: that addition in the embedding space is what encodes the composition of meaning. Apart from the random baseline, we further carry out experiments to check the compositionality captured in different layers of SBERT (page7), different training stages of BERT (page 8), and different training stages of knowledge graph embedding (page 10). This can serve as other baselines.

---

> > > ### Comment · Reviewer_yF7A · 2025-04-02
> > > **Thank you for your rebuttal**
> > >
> > > I sincerely thank the authors for taking the time to reply to my review and for the new changes.  After reading your well-written and insightful rebuttal, together with the reviews and discussions elsewhere in this submission, I still have some concerns regarding the comparison of this work to that of Xu et al. (2023) [3] and the depth of the provided analysis (particularly its actionability). As my main concerns may require significant changes in the current manuscript to address if I am correctly interpreting the contributions, I believe it is best to outline my doubts below and let the AE make an informed decision for this work considering this and my fellow reviewers' comments.
> > >
> > > > Figure 2 + Lack of error bars on tables and figures
> > >
> > > I understand what a p-value is and could see the benefit of showing the entire distribution as an extra addition. However, I am still not convinced it is a very useful representation for the main body of the paper, given how much space it takes vs the value it brings. More importantly, however, **I can't really see why this is an argument for not including error bars on the statistics collected**, as it is dictated by conventional good scientific practice.
> > >
> > > > Commenting on the semantic space
> > >
> > > Thanks for this reply! The proposed change in narrative looks reasonable and better to me.
> > >
> > > > Intuition behind the (very) high real and random cosine similarities
> > >
> > > This makes sense. Thank you for clarifying this.
> > >
> > > > Changes + Improvements over Xu et al. (2023) [3]
> > >
> > > Thank you for replying to this concern. If I am honest, I am still on the fence about this, and I believe it would benefit from a discussion with other reviewers + the AE. Firstly, contrary to what is claimed in the rebuttal, I would not qualify this work as a reproducibility study of [3] since, above and in the paper itself, the authors do not claim this and instead claim that this is an extension/application of ideas in [3]. Second, I can see that the experiments are indeed applications of this previous framework, which required some minor extensions of it. However, my main concern here is that the experiments, although certainly new, are lacking depth or actionable/unexpected new insights. This last point is obviously subjective, and I am happy to be convinced otherwise by my fellow reviewers or the AE.
> > >
> > > > Lack of actionability
> > >
> > > I do sincerely understand the benefit of identifying and studying the behaviour of compositionality in different modes, training stages, etc. However, the question of how this leads to actionable changes one can do to improve some aspect of interest was not entirely addressed (particularly vis-a-vis already existing approaches such as, say, debiasing methods). For example, one may claim this is useful for mechanistic interpretability, but it is hard to take that at face value without comparing the discussed methodology to any of the dozens of methods in Mech Interp that attempt to discover and use similarly "composable" representations. As such, I can't immediately see what sort of actions one may take based on these results.
> > >
> > > > Use of a random baseline
> > >
> > > Studying compositionality across different stages, architectures, etc, is useful and important but **does not really address my concern regarding the use of a random baseline**. Very well-known and traditional methodologies such as dictionary learning, sparse autoencoders, topic learning, non-negative matrix factorization, etc, all attempt to answer similar questions in different fields. Therefore, there are a lot of other baselines one can adopt to evaluate the proposed methodology that are stronger than the very weak random baseline. More importantly, I would politely disagree with the characterization that studying compositionality across different stages, architectures, etc, "can serve as other baselines." Baselines are almost by definition different approaches to the proposed methodology. Therefore, in my opinion, seeing how the results the same method change across different setups are not valid "other baselines."

---

### Decision · Action_Editor_KRQn · 2025-04-11

**Recommendation:** Reject

**Comment:**

While reviewers shared positive comments on the work such as its significance (yF7A), the experiments (H4fb, GqpX), and correctness (GqpX), they also raised several concerns.

yF7A shared concerns on 3 major points: 1) the overlap with a previous work [a] and potential violation of the TMLR guidelines, 2) the lack of actionability of the findings, 3) the random baseline used as main reference for the experimental comparisons. H4fb disagrees with the terminology used and that the core claims relate to compositionality.  GqpX found issues in the clarity of the manuscript.

The response and updated manuscript partially addressed these concerns. While GqpX found the clarity improved, H4fb pointed still to issues in the terminology used, requiring further changes. Moreover, yF7A found their comments on the adherence to the TMLR policy not addressed, as well as the use of a random baseline as main reference (highlighted also by GqpX as a weakness).

The final updated manuscript lead to split recommendations, with two reviewers leaning toward a rejection and one toward acceptance.
The AE went through the work, the reviews, and the responses and found no evidence to overturn the majority. As pointed bout y the authors, the TMLR guidelines state that contributions that 1) "accounts of applications of existing techniques that shed light on the strengths and weaknesses of the methods," or 2) "reproducibility studies of previously published results or claims" are acceptable. However, the latter is not the case (as pointed out by yF7A) and, while the AE agrees with GqpX that "actionability" is not a requirement, they still consider the first type of contribution as questionable in the manuscript as it implies "shedding lights" on strengths and weaknesses but reviewers agree that the findings mostly confirm the application of [a] to more settings rather than being an in-depth analysis.

Considering the highlighted weaknesses, the AE recommends the rejection of the manuscript in its current version.

**References:**
[a] Xu, Zhaozhen, Zhijin Guo, and Nello Cristianini. "On compositionality in data embedding." International Symposium on Intelligent Data Analysis. Cham: Springer Nature Switzerland, 2023.

**Audience:**

Studying the compositionality of data embeddings is an interesting research topic that contributes to multiple directions and topics on machine learning, ranging from OOD generalization to debiasing.

**Claims And Evidence:**

The manuscript explores the compositionality of data embeddings, with the core claims being the additive compositionality hypothesis and its application to a variety of data embeddings (e.g., word, sentence, knowledge graph) and their composition as a linear composition of their parts (e.g., morphological information, words, attributes), studying this using two methods. The paper proposes two methods for detecting this i.e., correlation-based (studying the correlation between data and attributes) and additive (treating embeddings as the sum of meaningful attributes).

While generally satisfied by the width of the experiments, reviewers criticized the choice of the random permutation baseline and shared contrastive opinions on whether it is sufficient to provide enough statistical evidence to support the hypothesis. Moreover, they challenged the significance of this claim w.r.t. to previous contributions (see comments).

---

> ### Author Response · Authors · 2025-04-14
> **Clarification on Rejection Decision**
>
> Dear Editors In Chief, Action Editor, and Reviewers,
>
> Thank you for your time, review, and discussion. I accept the rejection decision; however, to improve this paper for future submissions, I would appreciate clarification on which specific claims in the submission were **NOT** supported by accurate, convincing, and clear evidence.
>
> I acknowledge that the paper is an application of existing methods to different types of embeddings.
>
> If the rejection is only based on the contribution is not enough and lack of novelty, I completely understand and happy to take it. Even though TMLR states that novelty is not a primary consideration and I personally believe finding compositionality across different types of data, different objective functions, different training stages is non-trivial.
>
> If the decision stems from the terminology we used, I find that reasoning unclear. Reviewer H4fb clearly made several incorrect points in the rebuttal, and I would be happy to de-anonymize and engage in further discussion to clarify.
>
> Regarding the random baseline, even the reviewer consider it weak, but it is the point we make for this paper (see comments made by reviewer GqpX). Demonstrating performance above a random baseline clearly shows the presence of a compositional signal, and it is precisely the main point of our paper (and this is the evidence of the paper!). If we take some other baselines such as dictionary learning or non-negative matrix factorization as suggested by Reviewer yF7A, we still need to do the random shuffling again to support our claim, since again this is the point of this paper. Also,
>
> If it is because of the overlap with a previous work, I don't believe it is the case... otherwise it should be desk rejected.
>
> Thank you again for your time and assistance. To improve this paper in the future, I would greatly appreciate it if you could let me know which specific claims are not supported by clear evidence.
>
> Many thanks,
>
> Authors (Can I de-anonymize at this point?)

---

> > ### Comment · Action_Editor_KRQn · 2025-04-15
> >
> > Dear Authors,
> >
> > Thank you for the feedback. The terminology and the baseline are elements for the decision but not core ones: they are aspects one may work on to further improve the work. The overlap with the previous work is a core element.
> >
> > In this regard, the overlap can also be spotted during the review phase (not only via a desk rejection), and I apologize if I did not flag this potential issue before. As the article extends previous work, other venues that welcome journal extensions may be a better fit than TMLR.
> >
> > Best regards,\
> > AE

---

> > > ### Author Response · Authors · 2025-04-15
> > > **Thank you for the feedback (Persistent Overlap?)**
> > >
> > > Dear Action Editor,
> > >
> > > Thank you for the feedback and thank you for coordinating this submission.
> > >
> > > Thanks again for all the reviews.
> > >
> > > Best wishes,
> > >
> > > Author
> > >
> > > PS1:  I don't think "overlap" accurately captures the situation, as we use established methods to produce new findings on distinct data types. However, if the AE considers it overlapping, then so be it.
> > >
> > > PS2: We assumed that being sent for review meant the previous overlap issue had been resolved. I personally feel a bit surprised that, after two submissions and extensive rebuttals and experiments, it was rejected again for the same reason.